# Igeood: An Information Geometry Approach to Out-of-Distribution Detection

**Eduardo D. C. Gomes, Florence Alberge & Pierre Duhamel**
Laboratoire des signaux et systèmes (L2S)
Université Paris-Saclay CNRS CentraleSupélec
91190, Gif-sur-Yvette, France.
`{eduardo.dadalto,florence.alberge,pierre.duhamel}@centralesupelec.fr`

**Pablo Piantanida**
International Laboratory on Learning Systems (ILLS)
McGill ETS MILA CNRS Université Paris-Saclay CentraleSupélec
H3C 1K3 Quebec, Canada
`piantani@mila.quebec`

## Abstract

Reliable out-of-distribution (OOD) detection is fundamental to implementing safer modern machine learning (ML) systems. In this paper, we introduce Igeood, an effective method for detecting OOD samples. Igeood applies to any pre-trained neural network, works under various degrees of access to the ML model, does not require OOD samples or assumptions on the OOD data but can also benefit (if available) from OOD samples. By building on the geodesic (Fisher-Rao) distance between the underlying data distributions, our discriminator can combine confidence scores from the logits outputs and the learned features of a deep neural network. Empirically, we show that Igeood outperforms competing state-of-the-art methods on a variety of network architectures and datasets.

## 1 Introduction

Deep neural networks (DNNs) reach the state-of-the-art in several classification tasks as they are known to generalize well on data with a distribution close to the training set. Whereas, in many practical applications, the training set does not reflect well enough the real-life environment (Quionero-Candela et al., 2009) which is often non-stationary and sometimes with unpredictable events. Therefore, matching the training scenario to reality can be impossible or too complex. The inability of machine learning (ML) models to adapt to non-stationary distributions could limit their adoption in mission-critical systems (e.g., autonomous devices, healthcare applications).

Out-of-Distribution (OOD) or novelty detection is one of the main objectives in conceiving reliable ML systems (Amodei et al., 2016). A typical application is monitoring ML-based online services for periodically shifting distributions. However, tracking changes in the underlying data distribution is challenging as they contain unusual (irregular or unexpected) events and have large dimensions. For instance, relying on the intrinsic properties of ML models and their statistical behavior in the presence of in-distribution data is essential to identify OOD samples. Classic approaches to OOD detection consist of deriving metrics for detecting those abnormalities from the lens of ML models (e.g., softmax output, latent representations across layers), provided that often only a single test example is available. Furthermore, these metrics are subject to potential limitations inherent in practical scenarios depending on the level of access to information in the ML model, e.g., having access only to the last layer or to all intermediate layers.

The baseline approach for OOD detection relies on the predictive uncertainty of DNNs. Hendrycks & Gimpel (2017) demonstrated that OOD samples, in general, induce DNN classifiers to output less confident softmax scores, while existing state-of-the-art methods on classification problems still output high accuracy even under dataset shift. For instance, Ovadia et al. (2019) show that as the accuracy of the underlying DNN increases, the supervisors' outlier detection accuracy also

improves. Unfortunately, also the variance increases. Henriksson et al. (2021) observed that small changes in model parameters that marginally impact the accuracy could have a degrading impact on the performance of the OOD discriminator. This challenge is not exclusive to discriminative models. Deep generative models also fall short in discerning OOD from in-distribution samples. Nalisnick et al. (2019) raise awareness of the fact that deep generative models also may output a higher likelihood to OOD samples. They show that, even though the samples from the in-distribution CIFAR-10 (Krizhevsky et al., 2009) dataset (e.g., cats, dogs, airplanes, ships) are conceptually and visually different from house numbers from SVHN (Netzer et al., 2011) dataset, DNN-based classifiers may still assign a high likelihood to SVHN samples.

In this paper, we propose IGEOOD, a new unified and effective method to perform OOD detection by rigorously exploring the information-geometric properties of the feature space on various depths of a DNN. IGEOOD provides a flexible framework that applies to any pre-trained softmax neural classifier. A key ingredient of IGEOOD is the Fisher-Rao distance. This distance is used as an effective differential geometry tool for clustering and as a distance in the context of multivariate Gaussian pdfs (Pinele et al., 2020; Strapasson et al., 2016). In our context, we measure the dissimilarity between probability distributions (in and out), as the length of the shortest path within the manifold induced by the underlying class of distributions (i.e., the softmax probabilities of the neural classifier or the densities modeling the learned representations across the layers). By doing so, we can explore statistical invariances of the geometric properties of the learned features (Bronstein et al., 2021). Our method adapts to the various scenarios depending on the level of information access of the DNN and uses only in-distribution samples but can also benefit (if available) from OOD samples.

## 1.1 CONTRIBUTIONS

Our work investigates the problem of OOD detection and advances state-of-the-art in different ways.

**i** To the best of our knowledge, this is the first work studying *information geometry* tools to devise a unified metric for OOD detection. We derive an explicit characterization of the Fisher-Rao distance based on the information-geometric properties of the softmax probabilities of the neural classifier and the class of multivariate Gaussian pdfs. In general terms, our Fisher-Rao-based metric measures the mismatch–in the geometry space–between the probability density functions of the pre-trained DNN classifier conditioned on test and in-distribution samples. Section 3 details IGEOOD.

**ii** Experiments on BLACK-BOX and GREY-BOX setups using various datasets, architectures, and classification tasks show that IGEOOD is competitive with state-of-the-art methods. In the BLACK-BOX setup, we assume that only the outputs, i.e., the logits of the DNN, are available. In the GREY-BOX setup, we allow access to all parameters of the network; however, the detection must be performed using only the output softmax probabilities. The latter permits input pre-processing which introduces a small (additive) noise in the direction of the gradients w.r.t the test sample. This pre-processing allows for further discrimination between in- and out-of-distribution samples. Our benchmark contains two DNN architectures, three in-distribution datasets, and nine OOD datasets.

**iii** In a WHITE-BOX setting, we combine the logits with the low-level features of the DNN to leverage further useful statistical information of the encoded in-distribution data. We model the pre-trained latent representations as a mixture of Gaussian pdfs with a diagonal covariance matrix. Under this assumption, we derive a confidence score based on the Fisher-Rao distance between conditional pdfs corresponding to the test and the closest in-distribution samples. Experiments based on various datasets, architectures, and classification tasks clearly show consistent improvement of IGEOOD, achieving new state-of-the-art performance on a couple of benchmarks. In particular, we increased the average TNR at 95% TPR by 11.2% with tuning on OOD data and by 2.5% with tuning on adversarial data compared to Lee et al. (2018).

## 1.2 RELATED WORKS

OOD discriminators consist of a binary classifier to distinguish between in- and out-of-distribution samples. A few works (Shalev et al., 2018; Hendrycks et al., 2019; Bitterwolf et al., 2020; Mohseni et al., 2020; Winkens et al., 2020; Vyas et al., 2018; Hein et al., 2019) propose retraining the base (or an auxiliary) model with synthetic or ground truth OOD samples to serve as a classifier and as an OOD discriminator. Disposing of both OOD and in-distribution samples during training en-

ables the latent representations to learn the decision boundaries to facilitate OOD detection. These methods will not be compared to ours in this work, as they entail retraining or modifying the base neural network by using OOD data to further train parameters. Nagarajan et al. (2021) studies failure modes of OOD detection methods to better understand how to improve them, especially how spurious features like the background can vastly degrade detection performance. Lee et al. (2021) leverage OOD data as a regularization technique to improve the generalization and robustness of current neural networks. References (Schlegl et al., 2017; Kirichenko et al., 2020; Choi & Jang, 2018; Vernekar et al., 2019; Xiao et al., 2020; Ren et al., 2019; Zhang et al., 2021; Mahmood et al., 2021; Zhang et al., 2020; Zisselman & Tamar, 2020) study OOD detection in the context of generative models for density estimation. Open set recognition (Bendale & Boult, 2016), outlier or anomaly detection (Pimentel et al., 2014), concept drift detection (Quionero-Candela et al., 2009), and adversarial attacks detection (Goodfellow et al., 2015; Madry et al., 2018) are related topics.

**BLACK-BOX and GREY-BOX scenarios.** It is often the case on ML as a service (Ribeiro et al., 2015) that the model's parameters knowledge and access are not allowed to the end-user, granting access only to the computation of the forward and the logits or softmax outputs. The baseline work (Hendrycks & Gimpel, 2017) for BLACK-BOX techniques simply consider the unscaled maximum value of the softmax (MSP) as OOD score. In some cases, this confidence score is enough to distinguish between in-distribution and out-of-distribution examples, but it also may assign overconfident values to OOD examples (Hein et al., 2019). ODIN's (Liang et al., 2018) method has two variations. The BLACK-BOX variation consists of temperature scaling the softmax outputs. While the GREY-BOX variation also uses an input pre-processing technique that calculates the gradient of the model parameters and adds to the input in an adversarial manner for a more effective OOD detection. Hsu et al. (2020) proposes a variation of ODIN that does not need access to OOD data for validation. Liu et al. (2020) proposes an energy-based OOD score. They substitute the softmax confidence score with the free energy function with a temperature parameter without retraining. They also propose a GREY-BOX variation with posterior processing for improved results. Fine-tuning is done differently across the literature and should be considered when comparing methodologies.

**WHITE-BOX scenario.** This class of OOD detectors has access to all intermediate layer outputs. Naturally, discriminators have access to more information than the BLACK-BOX or GREY-BOX setups, warranting greater detection capacity. Batch-normalization statistics between layers are used (Quintanilha et al., 2019) to fit a logistic regression that serves as an OOD detection score. Sastry & Oore (2020) proposes high order Gram matrices to perform OOD detection by computing class-conditional pairwise feature correlations between the test sample and the training set across the hidden layers of the network. Lee et al. (2018) assume that latent features of DNN models trained under the softmax score follow a class-conditional Gaussian mixture distribution with tied covariance matrix and different class-conditional mean vectors. They calculate the Mahalanobis distance between a test sample as a single estimator of the mean of a class-conditional Gaussian distribution with a tied covariance matrix estimated on the training set. The importance of each low-level component and hyperparameters are tuned using validation data. Ren et al. (2021) modifies this method to improve detection of near-OOD data. They fit the layer-wise background distribution with a Gaussian distribution fit from the training set. They subtract the Mahalanobis distance between the test example and this distribution from the score proposed in Lee et al. (2018), reducing the importance of features shared by in- and out-of-distribution data.

## 2 BACKGROUND

Let $\mathcal{X} \subseteq \mathbb{R}^d$ be the feature space (continuous) and $\mathcal{Y}$ a label space. Moreover, let $p_{XY}$ be the underlying unknown probability density function (pdf) over $\mathcal{X} \times \mathcal{Y}$. We define the *in-distribution training dataset* as $\mathcal{D}_N \triangleq \left\{ (\boldsymbol{x}_i, y_i) \right\}_{i=1}^{N} \sim p_{XY}$, where $\boldsymbol{x}_i \in \mathcal{X}$ is the input feature data, $y_i \in \mathcal{Y} \triangleq \{1, \ldots, C\}$ is the output class among $C$ possible classes and $N$ denotes the number of training samples. The training dataset is characterized by the joint pdf $p_{XY}$ with *in-distribution marginals* $X \sim p_X$ and $Y \sim P_Y$. The predictor denoted by $f_{\mathcal{D}_N} : \mathcal{X} \to \mathcal{Y}$ is based on the inferred model $P_{\widehat{Y}|X}$, i.e., $f_{\mathcal{D}_N}(\boldsymbol{x}) \equiv f_n(\boldsymbol{x}; \mathcal{D}_N) \triangleq \arg\max_{y \in \mathcal{Y}} P_{\widehat{Y}|X}(y|\boldsymbol{x}; \mathcal{D}_N)$. In order to model the underlying problem, we introduce an artificial binary random variable $Z \in \{0, 1\}$ indicating with $z = 1$ that the test sample $\boldsymbol{x}$ is OOD and otherwise, it is in-distribution. The open-world data can then be modeled as a *mixture* distribution $p_{X|Z}$ defined by $p_{X|Z}(\boldsymbol{x}|z = 0) \triangleq$

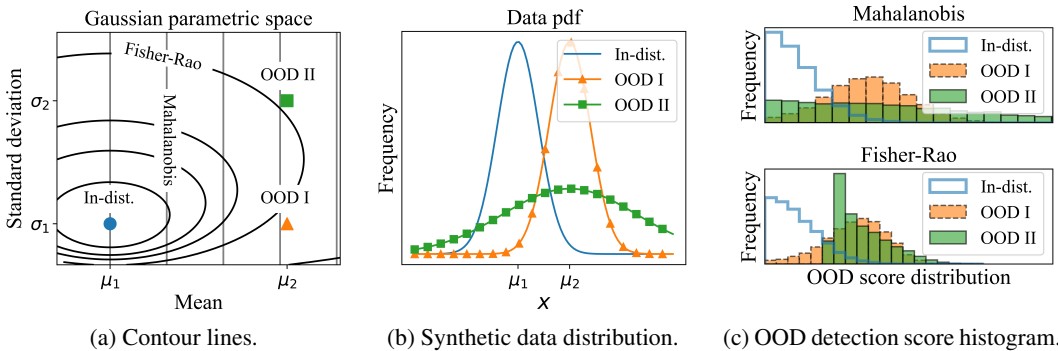

(a) Contour lines.          (b) Synthetic data distribution.          (c) OOD detection score histogram.

Figure 1: Example comparing Fisher-Rao with Mahalanobis distances to distinguish between 1D Gaussian distributions, showcasing the motivation to use of Fisher-Rao metric for OOD detection.

$p_X(\boldsymbol{x})$, and $p_{X|Z}(\boldsymbol{x}|z = 1) \triangleq q_X(\boldsymbol{x})$. The intrinsic difficulty arises from the fact that very little can be assumed about the unknown distributions $p_X$ and $q_X$, in particular for out-of-distribution.

## 3  IGEOOD: OOD DETECTION USING THE FISHER-RAO DISTANCE

This section introduces IGEOOD, a flexible framework for OOD detection. IGEOOD is implemented in two ways: at the level of the logits using temperature scaling (Section 3.2), which mitigates the high-confidence scores assigned to OOD examples, and layer-wise level (Section 3.3). The key ingredient of IGEOOD is the Fisher-Rao distance that allows for effective differentiation between in-distribution and out-of-distribution samples. This distance measures the dissimilarity between two probability models within a class of probability distributions by calculating the geodesic distance between two points on the learned manifold. This measure connects information geometry and differential geometry through the R. Fisher information matrix (Fisher, 1922). Closed-form expressions of this distance are known to multivariate normal distributions under certain assumptions, among others distributions (Pinele et al., 2020).

### 3.1  MOTIVATION FOR THE USE OF THE FISHER-RAO DISTANCE FOR OOD DETECTION

We introduce a simple example to demonstrate conceptually how Fisher-Rao distance is instrumental to OOD detection. It should be noted that this example is limited to one dimension. However, we expect similar behavior with more complex data under the Gaussianity assumptions.

Consider the case where we try to distinguish between samples from distinct Gaussian distributions on 1D. Assume that the in-distribution data follows a Gaussian $\mathcal{N}(\mu_1, \sigma_1)$ while OOD data is drawn according to either $\mathcal{N}(\mu_2, \sigma_1)$ or $\mathcal{N}(\mu_2, \sigma_2)$. These distributions are illustrated in Figures 1a and 1b. In this setup, distance-based approaches which are invariant to the variance of the distributions would have the performance limited to the information given by the difference between the means of the underlying distributions. For instance, in the case of the Mahalanobis distance, we would rely our discrimination on the difference between the sample and the in-distribution mean, rescaled by the in-distribution standard deviation only, but nothing further could be obtained. However, if we can estimate OOD standard deviations from actual or pseudo OOD data, we expect the Fisher-Rao distance between Gaussian distributions to be more effective in distinguishing between distributions. Figure 1c shows that the Fisher-Rao distance distinguishes better between "In-dist." and "OOD II" samples, while the other distances fail.

### 3.2  IGEOOD SCORE USING THE SOFTMAX PROBABILITY

The Fisher-Rao distance (Atkinson & Mitchell, 1981) takes as input two probability distributions. For the classification problem, we can take the temperature $T$ scaled softmax function (Eq. (1)) as

an approximation of a class-conditional probability distribution:

$$q_{\boldsymbol{\theta}}\left(y|f(\boldsymbol{x}); T\right) \triangleq \frac{\exp\left(f_y(\boldsymbol{x})/T\right)}{\sum_{y' \in \mathcal{Y}} \exp\left(f_{y'}(\boldsymbol{x})/T\right)}, \tag{1}$$

where $f : \mathcal{X} \to \mathbb{R}^C$ is a vectorial function with $f \triangleq \left(f_1, f_2, \ldots, f_C\right)$ and $f_y(\cdot)$ denotes the $y$-th logits output value of the DNN classifier. The Fisher-Rao distance $d_{\mathrm{FR-Logits}}$ between two distributions resulting from the softmax probability evaluated at two data points is (see Appendix A):

$$d_{\mathrm{FR-Logits}}\left(q_{\boldsymbol{\theta}}(\cdot|f(\boldsymbol{x})), q_{\boldsymbol{\theta}}(\cdot|f(\boldsymbol{x}'))\right) \triangleq 2\arccos\left(\sum_{y \in \mathcal{Y}} \sqrt{q_{\boldsymbol{\theta}}\left(y|f(\boldsymbol{x})\right)q_{\boldsymbol{\theta}}\left(y|f(\boldsymbol{x}')\right)}\right). \tag{2}$$

**Class conditional centroid estimation.** We model the training dataset class-conditional posterior distribution by calculating the centroid of the logits representations of this set. Precisely, we compute the *empirical centroid* for the logits of each class $y \in \mathcal{Y} = \{1, \ldots, C\}$ of the in-distribution training dataset $\mathcal{D}_N$ corresponding to the Fisher-Rao distance, i.e.,

$$\boldsymbol{\mu}_y \triangleq \min_{\boldsymbol{\mu} \in \mathbb{R}^C} \frac{1}{N_y} \sum_{\forall\, i\,:\, y_i = y} d_{\mathrm{FR-Logits}}\left(q_{\boldsymbol{\theta}}(\cdot|f(\boldsymbol{x}_i)), q_{\boldsymbol{\theta}}(\cdot|\boldsymbol{\mu})\right), \tag{3}$$

where $N_y$ is the amount of training examples with label $y$. We optimize this expression offline using SGD algorithm, where the parameter to be tuned is $\boldsymbol{\mu}$ in the logits space. This is equivalent to finding the centroid of a cluster using the Fisher-Rao distance, after each example has been assigned to a cluster. Please refer to the appendix (see Section B) for further details on this optimization.

**OOD and confidence score.** Using the softmax probability, we can define a confidence score to be the minimum of the Fisher-Rao distance between $f(\boldsymbol{x})$ and the class-conditional centroids. As a sanity check, we show empirically in the appendix (see Section C) that this confidence score does not degrade the in-distribution test classification accuracy. Thus, the estimated class $\widehat{y}_{\mathrm{FR}}$ follows as:

$$\widehat{y}_{\mathrm{FR}}(\boldsymbol{x}) \triangleq \arg\min_{y \in \mathcal{Y}} d_{\mathrm{FR-Logits}}\left(q_{\boldsymbol{\theta}}(\cdot|f(\boldsymbol{x})), q_{\boldsymbol{\theta}}(\cdot|\boldsymbol{\mu}_y)\right). \tag{4}$$

However, we obtained slightly better OOD detection performance by using Eq. (5) instead of the minimal value. A likely explanation would be that this metric uses extra information from the other logits dimensions. We provide an empirical study comparing both methods in the appendix (see Section E.1). Thus, we propose the Fisher-Rao distance-based OOD detection score $\mathrm{FR}_0(\boldsymbol{x})$ for the logits to be the sum of the distances between $f(\boldsymbol{x})$ and each individual class conditional centroid $\boldsymbol{\mu}_y$ given by Eq. (3). By taking the sum instead of the minimal distance, we leverage useful information related to the example's confidence score for each class $y$. We denote it by

$$\mathrm{FR}_0(\boldsymbol{x}) \triangleq \sum_{y \in \mathcal{Y}} d_{\mathrm{FR-Logits}}\left(q_{\boldsymbol{\theta}}(\cdot|f(\boldsymbol{x})), q_{\boldsymbol{\theta}}(\cdot|\boldsymbol{\mu}_y)\right). \tag{5}$$

**Input pre-processing.** In consonance with the literature (Liang et al., 2018; Liu et al., 2020; Lee et al., 2018), we also perform input pre-processing to enhance the detection between in-distribution and OOD samples and potentially improve OOD detection performance for the GREY-BOX discriminator. We add small magnitude perturbations $\varepsilon$ in a Fast Gradient-Sign Method-style (FGSM) (Goodfellow et al., 2015) to each test sample $\boldsymbol{x}$ to increase the proposed metric, that is:

$$\widetilde{\boldsymbol{x}} = \boldsymbol{x} + \varepsilon \odot \mathrm{sign}\left[\nabla_{\boldsymbol{x}} \mathrm{FR}_0(\boldsymbol{x})\right]. \tag{6}$$

**The OOD detector.** The detector consists of a threshold-based function for discriminating between in-distribution and OOD data. This threshold $\delta$ and parameters are set so that the true positive rate, i.e., the in-distribution samples correctly classified as in-distribution, becomes 95%. Mathematically, the BLACK-BOX OOD detector $g_{\mathrm{BB}}$ and the GREY-BOX OOD detector $g_{\mathrm{GB}}$ writes:

$$g_{\mathrm{BB}}(\boldsymbol{x}; \delta, T) = \begin{cases} 1 & \text{if } \mathrm{FR}_0(\boldsymbol{x}) \leq \delta \\ 0 & \text{if } \mathrm{FR}_0(\boldsymbol{x}) > \delta \end{cases} \quad \text{and} \quad g_{\mathrm{GB}}(\widetilde{\boldsymbol{x}}; \delta, T, \varepsilon) = \begin{cases} 1 & \text{if } \mathrm{FR}_0(\widetilde{\boldsymbol{x}}) \leq \delta \\ 0 & \text{if } \mathrm{FR}_0(\widetilde{\boldsymbol{x}}) > \delta \end{cases}. \tag{7}$$

### 3.3 IGEOOD SCORE LEVERAGING LATENT FEATURES

For each layer, we define a set of class-conditional Gaussian distributions with diagonal standard deviation matrix $\boldsymbol{\sigma}^{(\ell)}$ and class-conditional mean $\boldsymbol{\mu}_y^{(\ell)}$, where $y \in \{1, \ldots, C\}$ and $\ell$ is the index of the latent feature. We compute the empirical estimates of these parameters according to

$$\boldsymbol{\mu}_y^{(\ell)} = \frac{1}{N_y} \sum_{\forall i : y_i = y} f^{(\ell)}(\boldsymbol{x}_i), \quad \text{and} \quad \boldsymbol{\sigma}^{(\ell)} = \mathrm{diag}\left(\sqrt{\frac{1}{N} \sum_{y \in \mathcal{Y}} \sum_{\forall i : y_i = y} \left(f_j^{(\ell)}(\boldsymbol{x}_i) - \mu_{y,j}^{(\ell)}\right)^2}\right), \quad (8)$$

where $j \in \{1, \ldots, k\}$, $k$ is the size of feature $\ell$, and $f^{(\ell)}(\cdot)$ is the output of the network for feature $\ell$. The Fisher-Rao distance $\rho_{\mathrm{FR}}$ between two arbitrary *univariate* Gaussian pdfs $\mathcal{N}(\mu_1, \sigma_1^2)$ and $\mathcal{N}(\mu_2, \sigma_2^2)$ is given by (See Section A)

$$\rho_{\mathrm{FR}}\left((\mu_1, \sigma_1), (\mu_2, \sigma_2)\right) = \sqrt{2} \log \frac{\left|\left(\frac{\mu_1}{\sqrt{2}}, \sigma_1\right) - \left(\frac{\mu_2}{\sqrt{2}}, -\sigma_2\right)\right| + \left|\left(\frac{\mu_1}{\sqrt{2}}, \sigma_1\right) - \left(\frac{\mu_2}{\sqrt{2}}, \sigma_2\right)\right|}{\left|\left(\frac{\mu_1}{\sqrt{2}}, \sigma_1\right) - \left(\frac{\mu_2}{\sqrt{2}}, -\sigma_2\right)\right| - \left|\left(\frac{\mu_1}{\sqrt{2}}, \sigma_1\right) - \left(\frac{\mu_2}{\sqrt{2}}, \sigma_2\right)\right|}. \quad (9)$$

Similarly, the Fisher-Rao distance $d_{\mathrm{FR-Gauss}}$ between two *multivariate* Gaussian pdfs with diagonal standard deviation matrix is derived from the univariate case and is given by

$$d_{\mathrm{FR-Gauss}}\left((\boldsymbol{\mu}, \boldsymbol{\sigma}), (\boldsymbol{\mu}', \boldsymbol{\sigma}')\right) = \sqrt{\sum_{i=1}^{k} \rho_{\mathrm{FR}}\left((\mu_i, \sigma_{i,i}), (\mu_i', \sigma_{i,i}')\right)^2}, \quad (10)$$

where $k$ is the cardinality of the distributions $\mathcal{N}(\boldsymbol{\mu}, \boldsymbol{\sigma})$ and $\mathcal{N}(\boldsymbol{\mu}', \boldsymbol{\sigma}')$, $\mu_i$ is the $i$-th component of the vector $\boldsymbol{\mu}$, and $\sigma_{i,i}$ is the entry with index $(i, i)$ of the standard deviation matrix $\boldsymbol{\sigma}$.

**Experimental support for a diagonal Gaussian mixture model.** It is known that intermediate features of a DNN can be valuable for detecting abnormal samples as demonstrated by Lee et al. (2018). Nonetheless, we observed that the latent features covariance matrices are often *ill-conditioned* and are diagonal dominant. In other words, the condition number of the covariance matrix often diverges, and the magnitude of the diagonal entry in a row is greater than or equal to the sum of all the other entries in that row for most rows. Thus, a diagonal covariance matrix will be a favorable compromise for OOD detection. See Appendix, Section B.3 for further details.

**Fisher-Rao distance-based feature-wise confidence score.** We derive a confidence score by applying the Fisher-Rao distance between the test sample $\boldsymbol{x}$ and the closest class-conditional diagonal Gaussian distribution. Contrarily to the logits, taking the sum did not improve results, so we kept the minimal distance. We can consider two scenarios: **(i)** We do not have access to any validation OOD data whatsoever. In this case, the natural choice is to model the test samples as Gaussian distribution with the same diagonal standard deviation as the learned representation, i.e.,

$$\mathrm{FR}_\ell(\boldsymbol{x}) = \min_{y \in \mathcal{Y}} d_{\mathrm{FR-Gauss}}\left((\boldsymbol{x}, \boldsymbol{\sigma}^{(\ell)}), (\boldsymbol{\mu}_y^{(\ell)}, \boldsymbol{\sigma}^{(\ell)})\right); \quad (11)$$

and **(ii)** we dispose of a validation OOD dataset on which the features' diagonal standard deviation matrices $\boldsymbol{\sigma}'^{(\ell)}$ and the means $\boldsymbol{\mu}'^{(\ell)}$ can be estimated, as well as the quantity:

$$\mathrm{FR}_\ell'(\boldsymbol{x}) = \min_{y \in \mathcal{Y}} d_{\mathrm{FR-Gauss}}\left((\boldsymbol{x}, \boldsymbol{\sigma}^{(\ell)}), (\boldsymbol{\mu}'^{(\ell)}, \boldsymbol{\sigma}'^{(\ell)})\right). \quad (12)$$

This validation dataset could be obtained from a synthetic dataset, a dataset different from the testing one, or even by adversarially creating OOD data by attacking the classifier model on the training dataset. In the appendix (Section B), we include pseudo-codes for calculating the IGEOOD score for the BLACK-BOX, GREY-BOX, and WHITE-BOX settings.

**Feature ensemble.** To further improve performance, we combine the confidence scores of the logits and the ones from the low-level features through a linear combination. Similarly to the strategy in Lee et al. (2018), we choose the weights $\alpha_0$, $\alpha_\ell$ and $\alpha_\ell' \in \mathbb{R}$ by training a logistic regression detector using validation samples. Thus, we ensure that the metric emphasizes features that demonstrate a greater capacity for detecting abnormal samples. IGEOOD score for the WHITE-BOX setting is:

$$\mathrm{FR}(\boldsymbol{x}) \triangleq \alpha_0 \mathrm{FR}_0(\boldsymbol{x}) + \sum_\ell \alpha_\ell \cdot \mathrm{FR}_\ell(\boldsymbol{x}) + \alpha_\ell' \cdot \mathrm{FR}_\ell'(\boldsymbol{x}), \quad (13)$$

where $FR_0$ is given by equation (5), $FR_\ell$ is given by equation (11) and $FR'$ considers a different validation diagonal covariance matrix for the test samples (equation (12)). We also apply input pre-processing similarly to the GREY-BOX setting (equation (6)), obtaining $FR(\widetilde{x})$ as final score.

**Unified metric.** For the three settings, the metric is the same but has different formulations given the family of the distributions. For the DNN outputs, we use the softmax posterior probability distribution formulation. For the intermediate layers, it is under the model of diagonal Gaussian pdfs. *Therefore, we have derived a unified OOD detection framework that combines a single distance for both the softmax outputs and the latent features of a neural network.* Figure 2 illustrates how each of the presented techniques contributes towards separating in-distribution and OOD samples. Additional histograms of the detection scores are relegated to the appendix (see Section F).

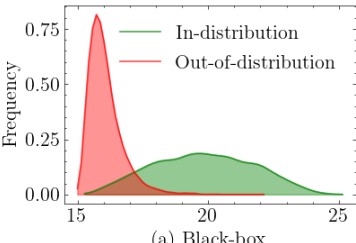 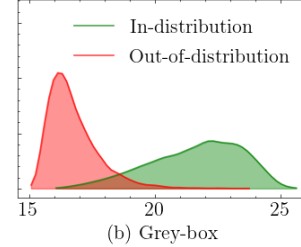 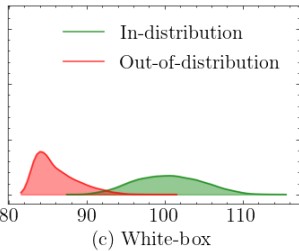

(a) Black-box        (b) Grey-box        (c) White-box

Figure 2: Probability distributions of the IGEOOD score under three different settings for a pre-trained DenseNet on CIFAR-10 for in-distribution and OOD data (TinyImageNet downsampled).

## 4 EXPERIMENTAL RESULTS

We show the effectiveness of IGEOOD comparing to state-of-the-art methods. Details about the experimental setup [1] and additional results are given in appendices (see Sections C, D, and E).

### 4.1 SETUP

The experimental setup follows the setting established by Hendrycks & Gimpel (2017), Liang et al. (2018) and Lee et al. (2018). We use two *pre-trained* deep neural networks architectures for image classification tasks: a Dense Convolutional Network (DenseNet-BC-100) (Huang et al., 2017) and a Residual Neural Network (ResNet-34) (He et al., 2016). We take as *in-distribution data* images from CIFAR-10 (Krizhevsky et al., 2009), CIFAR-100 and SVHN (Netzer et al., 2011) datasets.

For *out-of-distribution data*, we use natural image examples from the datasets: Tiny-ImageNet (Le & Yang, 2015), LSUN (Yu et al., 2015), Describable Textures Dataset (Cimpoi et al., 2014), Chars74K (de Campos et al., 2009), Places365 (Zhou et al., 2017), iSUN (Xu et al., 2015) and a synthetic dataset generated from Gaussian noise. For models pre-trained on CIFAR-10, data from CIFAR-100 and SVHN are also considered OOD; for models pre-trained on CIFAR-100, data from CIFAR-10 and SVHN are considered OOD, and for models pre-trained on SVHN, CIFAR-10 and CIFAR-100 datasets are considered OOD. We resize the images to dimension $32 \times 32$ by downsampling and applying center crop when needed. We only use test data for evaluation. Even though we ran experiments with image data, IGEOOD could be applied to any neural-based classification task.

We measure the effectiveness of the OOD detectors with three standard *evaluation metrics*: (i) The true negative rate at 95% true positive rate (TNR at TPR-95%); (ii) the area under the receiving operating curve (AUROC); and (iii) the area under the precision-recall curve (AUPR). We use the scores over the test set of in-distribution and OOD datasets to calculate them. For the BLACK-BOX and GREY-BOX experimental settings, we *tune hyperparameters* for all of the OOD detectors only based on the DNN classifier architecture, the in-distribution dataset, and a validation dataset. The iSUN (Xu et al., 2015) dataset is chosen as a source of OOD validation data, independently from OOD test data. We choose the parameters that maximize the TNR at TPR-95% on the validation OOD dataset. For the WHITE-BOX framework, we allow both the benchmark and our method to

---
[1]Our code is publicly available at `https://github.com/edadaltocg/Igeood`.

tune either on adversarially generated data from in-distribution training samples or a separate validation dataset containing $1,000$ images from the OOD test dataset with feature ensemble described in Section 3.3. In this case, we evaluate performance on the remaining test samples.

## 4.2 RESULTS FOR THE BLACK-BOX AND THE GREY-BOX SETUPS

For comparing IGEOOD under the hypothesis of a BLACK-BOX scenario, we consider the Baseline (Hendrycks & Gimpel, 2017) method, ODIN (Liang et al., 2018) with temperature scaling only, and the free-energy-based metric (Liu et al., 2020) with temperature scaling only. The results for the BLACK-BOX setting are available in Table 1, where we show the average and one standard deviation OOD detection performance for each of the eight OOD detection method in six different image classification contexts (couple DNN model and in-distribution dataset). The extended results for each OOD dataset can be found in Table 13. For comparison under the GREY-BOX assumption, we consider ODIN and the free-energy-based methods, both with input pre-processing. The results for the GREY-BOX setup are provided in the appendix (see Section E and Table 10). For the BLACK-BOX setting, IGEOOD slight improves the benchmark by less than 1% in TNR at TPR-95%. While for the GREY-BOX setting, results show IGEOOD is outperformed by <1% in a few benchmarks by ODIN, which is greatly improved by input pre-processing techniques.

Table 1: Average and standard deviation OOD detection performance across eight OOD datasets for each model and in-distribution dataset in a BLACK-BOX setting. IGEOOD is compared to Baseline (Hendrycks & Gimpel, 2017), ODIN (Liang et al., 2018), and Energy (Liu et al., 2020) methods. The extended results can be found in Table 13 in the appendix.

| Model | In-dist. | TNR at TPR-95% | AUROC |
|---|---|---|---|
| | | Baseline / ODIN / Energy / IGEOOD (ours) | |
| DenseNet | C-10 | $52.5_{\pm16}$/**$66.8_{\pm20}$**/$65.3_{\pm23}$/$65.6_{\pm23}$ | $91.8_{\pm3.2}$/**$92.8_{\pm4.6}$**/$92.1_{\pm5.3}$/$92.3_{\pm5.1}$ |
| | C-100 | $15.9_{\pm6.8}$/$20.5_{\pm9.5}$/$20.3_{\pm9.6}$/**$20.7_{\pm9.8}$** | $69.1_{\pm15}$/$71.6_{\pm20}$/$71.6_{\pm20}$/**$73.2_{\pm17}$** |
| | SVHN | $68.4_{\pm14}$/$68.8_{\pm20}$/$70.2_{\pm17}$/**$72.1_{\pm15}$** | **$92.3_{\pm4.0}$**/$87.3_{\pm14}$/$90.1_{\pm5.9}$/$90.9_{\pm5.3}$ |
| ResNet | C-10 | $41.7_{\pm16}$/$51.9_{\pm15}$/$56.3_{\pm13}$/**$56.7_{\pm13}$** | $89.6_{\pm3.1}$/$90.4_{\pm3.1}$/$90.4_{\pm3.0}$/**$90.5_{\pm3.0}$** |
| | C-100 | $15.0_{\pm5.5}$/$16.0_{\pm6.3}$/$16.3_{\pm7.1}$/**$16.4_{\pm6.8}$** | $74.0_{\pm1.9}$/$75.2_{\pm1.7}$/**$75.5_{\pm1.9}$**/**$75.5_{\pm1.7}$** |
| | SVHN | $76.2_{\pm7.8}$/$77.7_{\pm7.9}$/$78.0_{\pm7.9}$/**$78.3_{\pm8.0}$** | $92.2_{\pm2.9}$/$91.4_{\pm3.2}$/$91.4_{\pm3.2}$/$91.7_{\pm3.2}$ |
| Average and Std. | | $44.9_{\pm24}$/$50.3_{\pm24}$/$51.1_{\pm24}$/**$51.6_{\pm24}$** | $84.8_{\pm9.5}$/$84.8_{\pm8.3}$/$85.2_{\pm8.4}$/**$85.7_{\pm8.0}$** |

**Temperature scaling and input pre-processing.** We observed that low values of temperature and moderate noise magnitude yield better detection performance for IGEOOD on the logits. For most models and datasets, we obtained better results for temperatures between 1 and 6 and noise magnitudes below 0.002. Detailed results and the best hyperparameters found for each configuration, as well as figures of their impact on performance, are delegated to the appendix (see Section E).

**How the choice of validation dataset impacts performance.** We include in the appendix (see Section E) the average OOD detection performance for each method when we change the validation set among the nine available ones. We show that the average TNR at TPR-95% for IGEOOD ranges between 63% and 72% on a BLACK-BOX scenario and between 65% and 74% on a GREY-BOX scenario. The performances among the compared methods are consistent across validation datasets.

## 4.3 RESULTS FOR THE WHITE-BOX SETTING

For benchmarking IGEOOD on the WHITE-BOX setting, we compare results to the Mahalanobis (Lee et al., 2018) method with input pre-processing and feature ensemble. For both of them, we extract features from every output of the dense (or residual) block of the DenseNet (or ResNet) model and the first convolutional layer. The size of each feature is reduced by average pooling in the spatial dimensions. Thus, the initial dimension $\mathcal{F}_\ell \times \mathcal{W}_\ell \times \mathcal{H}_\ell$ is reduced to $\mathcal{F}_\ell$, where $\mathcal{F}_\ell$ is the number of channels in block $\ell$. For DenseNet, this reduction translates to features of sizes $\mathcal{F}_1 = \{24, 108, 150, 342\}$; and for ResNet, to features of sizes $\mathcal{F}_2 = \{64, 64, 128, 256, 512\}$.

We consider two scenarios for tuning hyperparameters for both Mahalanobis and IGEOOD: one with adversarially generated (FGSM) and in-distribution data and another one with 1,000 OOD samples

and in-distribution data. We derive two methods: IGEOOD+, which is given by equation (13) and considers that we can calculate the statistics from OOD data as additional information; and IGEOOD, which doesn't consider any prior on OOD data, i.e., set $\alpha'_\ell = 0$ on equation (13).

**Comparison with current literature.** For each DNN model and in-distribution dataset pair, we report the average and one standard deviation OOD detection performance for Mahalanobis (Lee et al., 2018), IGEOOD and IGEOOD+. Table 2 validates the contributions of our techniques. We observe substantial performance improvement in all experiments for the left-hand side of the table, where we outperform Mahalanobis on average for all test cases. IGEOOD+ show improvements of at least 2.1% up to 23% on TNR at TPR-95%. Since the results are usually above 90%, these improvements are significant. To assess the consistency of IGEOOD to the choice of validation data, we measured the detection performance when all hyperparameters are tuned only using in-distribution and generated adversarial data, as observed in the right-hand side of Table 2. IGEOOD record improvements up to 10.5%, and improves by 2.5% the average TNR at TPR-95% across all datasets and models. We provide an extra benchmark against other WHITE-BOX methods (Sastry & Oore, 2020; Hsu et al., 2020; Zisselman & Tamar, 2020) (see Table 11 in the appendix).

Table 2: Average and standard deviation OOD detection performance for the WHITE-BOX settings. The abbreviation TNR-95%, C-10 and C-100 stands for TNR at TPR-95%, CIFAR-10 and CIFAR-100, respectively. The extended results can be found in Tables 15 and 16 in the appendix.

| Model | In-dist. | Validation on OOD data | | Validation on adversarial data | |
|---|---|---|---|---|---|
| | | TNR-95% | AUROC | TNR-95% | AUROC |
| | | Mahalanobis / IGEOOD+ (ours) | | Mahalanobis / IGEOOD (ours) | |
| DenseNet | C-10 | $76.6_{\pm31}/\mathbf{92.6}_{\pm14}$ | $92.1_{\pm12}/\mathbf{98.4}_{\pm3.0}$ | $75.9_{\pm30}/\mathbf{77.9}_{\pm29}$ | $91.7_{\pm12}/\mathbf{94.0}_{\pm9.0}$ |
| | C-100 | $67.2_{\pm28}/\mathbf{90.2}_{\pm21}$ | $90.2_{\pm13}/\mathbf{97.7}_{\pm5.0}$ | $60.4_{\pm34}/\mathbf{70.9}_{\pm35}$ | $85.3_{\pm19}/\mathbf{90.8}_{\pm13}$ |
| | SVHN | $93.3_{\pm8.0}/\mathbf{98.0}_{\pm2.0}$ | $98.6_{\pm1.0}/\mathbf{99.6}_{\pm0.1}$ | $\mathbf{93.7}_{\pm10}/92.2_{\pm9.0}$ | $\mathbf{98.6}_{\pm2.0}/98.4_{\pm1.0}$ |
| ResNet | C-10 | $82.5_{\pm23}/\mathbf{91.6}_{\pm16}$ | $96.5_{\pm4.0}/\mathbf{98.4}_{\pm3.0}$ | $\mathbf{78.6}_{\pm24}/77.3_{\pm32}$ | $\mathbf{95.3}_{\pm6.0}/90.0_{\pm15}$ |
| | C-100 | $70.4_{\pm30}/\mathbf{86.4}_{\pm23}$ | $91.9_{\pm10}/\mathbf{97.1}_{\pm5.0}$ | $57.4_{\pm36}/\mathbf{65.1}_{\pm33}$ | $86.9_{\pm13}/\mathbf{88.6}_{\pm15}$ |
| | SVHN | $96.8_{\pm6.0}/\mathbf{98.9}_{\pm2.0}$ | $99.2_{\pm1.0}/\mathbf{99.7}_{\pm0.1}$ | $\mathbf{96.3}_{\pm8.0}/93.6_{\pm14}$ | $\mathbf{99.1}_{\pm1.0}/98.4_{\pm3.0}$ |
| Average and Std. | | $81.1_{\pm11}/\mathbf{92.9}_{\pm4.0}$ | $94.8_{\pm4.0}/\mathbf{98.5}_{\pm1.0}$ | $77.0_{\pm15}/\mathbf{79.5}_{\pm10}$ | $92.8_{\pm5.4}/\mathbf{93.4}_{\pm3.9}$ |

**Ablation study.** IGEOOD has three components, $FR_0$, $FR_\ell$, and $FR'_\ell$, that together compose the final metric of equation (13). The outputs of the network provide limited OOD detection capacity as observed in Table 1. When available, the intermediate features, i.e., $FR_\ell$, are a valuable resource for OOD detection. Moreover, when few reliable OOD data are available, calculating $FR'_\ell$ can further improve the detection performance (left-hand side column of Table 2). Also, data from a source other than in-distribution, e.g., adversarial samples, is enough for tuning hyperparameters and combining features (right-hand side column of Table 2). The detection capacity of each hidden layer before any tuning is studied in Appendix B.4. Experiments show that the Fisher-Rao metric effectively separates in- and out-of-distribution data for each of the features individually as well.

## 5 SUMMARY AND CONCLUDING REMARKS

This paper introduces IGEOOD, an effective and flexible method for OOD detection that applies to any pre-trained neural network. The main feature of IGEOOD relies on the geodesic distance of the probabilistic manifold of the learned latent representations that induces an effective measure for OOD detection. First, in a (GREY-) BLACK-BOX setup, we calculate the sum of the Fisher-Rao distance between the softmax output, corresponding to the test (pre-processed) sample, and a reference probability, corresponding to the conditional-class of softmax probabilities. Similarly, in a WHITE-BOX setup, we model the low-level features of a DNN as a diagonal Gaussian mixture. The Fisher-Rao distance between the pdf of the latent feature, corresponding to the test sample, and a reference pdf, corresponding to the conditional-class of pdfs, provides an effective confidence score. We considered diverse testing environments where prior knowledge of OOD data may or may not be available, reflecting diverse application scenarios. It is observed that IGEOOD significantly and consistently improves the accuracy of OOD detection on several DNN architectures across various datasets for a WHITE-BOX setting. Some perspectives for future work include studying causal factors, explainable components for OOD detection, and extensions to textual data.

ACKNOWLEDGMENTS

This work has been supported by the project PSPC AIDA: 2019-PSPC-09 funded by BPI-France.

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

# A    REVIEW OF FISHER-RAO DISTANCE (FRD)

In this section, we review some results from references Atkinson & Mitchell (1981); Pinele et al. (2020). We intend to clarify some basic concepts surrounding the Fisher-Rao distance while motivating this measure in the context of OOD detection.

In few words, the Fisher-Rao's distance is given by the geodesic distance, i.e., the shortest path between points in a Riemannian space induced by a parametric family. Consider the family $\mathcal{C}$ of probability distributions over the class of discrete concepts or labels: $\mathcal{Y} = \{1, \ldots, C\}$, denoted by $\mathcal{C} \triangleq \{q_{\boldsymbol{\theta}}(\cdot|\boldsymbol{x}) : \boldsymbol{x} \in \mathcal{X} \subseteq \mathbb{R}^C\}$.

We are interested in measuring the distance between probability distributions $q_{\boldsymbol{\theta}}(\cdot|\boldsymbol{x})$ with respect to the testing input $\boldsymbol{x}$ and a population of inputs drawn accordingly to the in-distribution data set. To this end, we first need to characterize the Fisher-Rao distance for two inputs or for two probability distributions $q_{\boldsymbol{\theta}}, q'_{\boldsymbol{\theta}} \in \mathcal{C}$.

Assume that the following regularity conditions hold (Atkinson & Mitchell, 1981):

(i)  $\nabla_{\boldsymbol{x}} q_{\boldsymbol{\theta}}(y|\boldsymbol{x})$ exists for all $\boldsymbol{x}, y$ and $\boldsymbol{\theta} \in \Theta$;

(ii)  $\sum_{y \in \mathcal{Y}} \nabla_{\boldsymbol{x}} q_{\boldsymbol{\theta}}(y|\boldsymbol{x}) = 0$ for all $\boldsymbol{x}$ and $\boldsymbol{\theta} \in \Theta$;

(iii)  $\boldsymbol{G}(\boldsymbol{x}) = \mathbb{E}_{Y \sim q_{\boldsymbol{\theta}}(\cdot|\boldsymbol{x})}\left[\nabla_{\boldsymbol{x}} \log q_{\boldsymbol{\theta}}(Y|\boldsymbol{x})\nabla_{\boldsymbol{x}}^{\top} \log q_{\boldsymbol{\theta}}(Y|\boldsymbol{x})\right]$ is positive definite for any $\boldsymbol{x}$ and $\boldsymbol{\theta} \in \Theta$.

Notice that if (i) holds, (ii) also holds immediately for discrete distributions over finite spaces (assuming that $\sum_{y \in \mathcal{Y}}$ and $\nabla_{\boldsymbol{x}}$ are interchangeable operations) as in our case. When (i)-(iii) are met, the variance of the differential form $\nabla_{\boldsymbol{x}}^{\top} \log q_{\boldsymbol{\theta}}(Y|\boldsymbol{x})d\boldsymbol{x}$ can be interpreted as the square of a differential arc length $ds^2$ in the space $\mathcal{C}$, which yields

$$ds^2 = \langle d\boldsymbol{x}, d\boldsymbol{x} \rangle_{\boldsymbol{G}(\boldsymbol{x})} = d\boldsymbol{x}^{\top} \boldsymbol{G}(\boldsymbol{x})d\boldsymbol{x}. \tag{14}$$

Thus, $\boldsymbol{G}$, which is the Fisher Information Matrix (FIM), can be adopted as a metric tensor. We now consider a curve $\boldsymbol{\gamma} : [0, 1] \to \mathcal{X}$ connecting a pair of arbitrary points $\boldsymbol{x}, \boldsymbol{x}'$ in the input space $\mathcal{X}$, i.e., $\boldsymbol{\gamma}(0) = \boldsymbol{x}$ and $\boldsymbol{\gamma}(1) = \boldsymbol{x}'$. Notice that any curve $\boldsymbol{\gamma}$ induces a curve $q_{\boldsymbol{\theta}}(\cdot|\boldsymbol{\gamma}(t))$ for $t \in [0, 1]$ in the space $\mathcal{C}$. The Fisher-Rao distance between the distributions $q_{\boldsymbol{\theta}} = q_{\boldsymbol{\theta}}(\cdot|\boldsymbol{x})$ and $q'_{\boldsymbol{\theta}} = q_{\boldsymbol{\theta}}(\cdot|\boldsymbol{x}')$ will be denoted as $d_{R,\mathcal{C}}(q_{\boldsymbol{\theta}}, q'_{\boldsymbol{\theta}})$ and is formally defined by the expression:

$$d_{R,\mathcal{C}}(q_{\boldsymbol{\theta}}, q'_{\boldsymbol{\theta}}) \triangleq \inf_{\boldsymbol{\gamma}} \int_0^1 \sqrt{\frac{d\boldsymbol{\gamma}^{\top}(t)}{dt}\boldsymbol{G}(\boldsymbol{\gamma}(t))\frac{d\boldsymbol{\gamma}(t)}{dt}}, \tag{15}$$

where the infimum is taken over all piecewise smooth curves. This means that the FRD is the length of the *geodesic* between points $\boldsymbol{x}$ and $\boldsymbol{x}'$ using the FIM as the metric tensor. In general, the minimization of the functional in equation (15) is a problem that can be solved using the well-known Euler-Lagrange differential equation.

## A.1    DERIVATION OF FISHER-RAO DISTANCE FOR THE CLASS OF SOFTMAX PROBABILITY DISTRIBUTIONS

The direct computation of the FIM of the family $\mathcal{C}$ with $q_{\boldsymbol{\theta}}(y|\boldsymbol{x})$ in the form of the softmax probability distribution function given by equation (1) can be shown to be singular, i.e., $\text{rank}(\boldsymbol{G}(\boldsymbol{x})) \leq C-1$, where $C-1$ is the number of degrees of freedom of the manifold $\mathcal{C}$. To overcome this issue, we introduce the probability simplex $\mathcal{P}$ defined by

$$\mathcal{P} = \left\{ q : \mathcal{Y} \to [0, 1]^C : \sum_{y \in \mathcal{Y}} q(y) = 1 \right\}. \tag{16}$$

Next, we consider the following parametrization for any distribution $q \in \mathcal{P}$:

$$q(y|\boldsymbol{z}) = \frac{z_y^2}{4}, \quad y \in \{1, \ldots, C\}. \tag{17}$$

From this expression, we consider the statistical manifold $\mathcal{D} = \left\{ q(\cdot|\boldsymbol{z}) : \|\boldsymbol{z}\|^2 = 4, z_y \geq 0, \forall y \in \mathcal{Y} \right\}$. Note that the parameter vector $\boldsymbol{z}$ belongs to the positive portion of a sphere of radius 2 and centered at the origin in $\mathbb{R}^C$. The computation of the FIM for $\boldsymbol{z}$ on $\mathcal{D}$ yields:

$$
\begin{aligned}
\boldsymbol{G}(\boldsymbol{z}) &= \mathbb{E}_{q(y|\boldsymbol{z})} \left[ \nabla_{\boldsymbol{z}} \log q(y|\boldsymbol{z}) \nabla_{\boldsymbol{z}}^\top \log q(y|\boldsymbol{z}) \right] \\
&= \sum_{y \in \mathcal{Y}} \frac{z_y^2}{4} \left( \frac{2}{z_y} \boldsymbol{e}_y \right) \left( \frac{2}{z_y} \boldsymbol{e}_y^\top \right) \\
&= \sum_{y \in \mathcal{Y}} \boldsymbol{e}_y \boldsymbol{e}_y^\top \\
&= \boldsymbol{I},
\end{aligned}
\tag{18}
$$

where $\{\boldsymbol{e}_y\}$ are the canonical basis vectors in $\mathbb{R}^C$ and $\boldsymbol{I}$ is the identity matrix. From equation (18) we can conclude that the Fisher-Rao metric in this parametric space is equal to the Euclidean metric. Also, since the parameter vector lies on a sphere, the FRD between the distributions $q = q(\cdot|\boldsymbol{z})$ and $q' = q(\cdot|\boldsymbol{z}')$ can be written as the radius of the sphere times the angle between the vectors $\boldsymbol{z}$ and $\boldsymbol{z}'$. Which leads to expression:

$$
d_{R,\mathcal{D}}(q, q') = 2 \arccos \left( \frac{\boldsymbol{z}^\top \boldsymbol{z}'}{4} \right) = 2 \arccos \left( \sum_{y \in \mathcal{Y}} \sqrt{q(y|\boldsymbol{z}) q(y|\boldsymbol{z}')} \right).
\tag{19}
$$

Finally, we can compute the FRD for softmax distributions in $\mathcal{C}$ as

$$
d_{\text{FR-Logits}}(q_{\boldsymbol{\theta}}, q'_{\boldsymbol{\theta}}) = 2 \arccos \left( \sum_{y \in \mathcal{Y}} \sqrt{q_{\boldsymbol{\theta}}(y|\boldsymbol{x}) q_{\boldsymbol{\theta}}(y|\boldsymbol{x}')} \right),
\tag{20}
$$

obtaining the same form of equation (2). Notice that $0 \leq d_{\text{FR-Logits}}(q_{\boldsymbol{\theta}}, q'_{\boldsymbol{\theta}}) \leq \pi$ for all $\boldsymbol{x}, \boldsymbol{x}' \in \mathcal{X} \subseteq \mathbb{R}^C$, being zero when $q_{\boldsymbol{\theta}}(\cdot|\boldsymbol{x}) = q_{\boldsymbol{\theta}}(\cdot|\boldsymbol{x}')$ and maximum when the vectors $\left( q_{\boldsymbol{\theta}}(1|\boldsymbol{x}), \ldots, q_{\boldsymbol{\theta}}(C|\boldsymbol{x}) \right)$ and $\left( q_{\boldsymbol{\theta}}(1|\boldsymbol{x}'), \ldots, q_{\boldsymbol{\theta}}(C|\boldsymbol{x}') \right)$ are orthogonal.

### A.2 DERIVATION OF FISHER-RAO DISTANCE FOR MULTIVARIATE GAUSSIAN DISTRIBUTIONS

Consider a broader statistical manifold $\mathcal{S} \triangleq \{p_{\boldsymbol{\theta}} = p(\boldsymbol{x}; \boldsymbol{\theta}) : \boldsymbol{\theta} = (\theta_1, \theta_2, \ldots, \theta_m) \in \Theta\}$ of multivariate differential probability density functions. The Fisher information matrix $\boldsymbol{G}(\boldsymbol{\theta}) = [g_{ij}(\boldsymbol{\theta})]$ in this parametric space is provided by:

$$
\begin{aligned}
g_{ij}(\boldsymbol{\theta}) &= \mathbb{E}_{\boldsymbol{\theta}} \left( \frac{\partial}{\partial \theta_i} \log p(\boldsymbol{x}; \boldsymbol{\theta}) \frac{\partial}{\partial \theta_j} \log p(\boldsymbol{x}; \boldsymbol{\theta}) \right) \\
&= \int \frac{\partial}{\partial \theta_i} \log p(\boldsymbol{x}; \boldsymbol{\theta}) \frac{\partial}{\partial \theta_j} \log p(\boldsymbol{x}; \boldsymbol{\theta}) p(\boldsymbol{x}; \boldsymbol{\theta}) dx.
\end{aligned}
\tag{21}
$$

Next, consider a multivariate Gaussian distribution:

$$
p(\boldsymbol{x}; \boldsymbol{\mu}, \Sigma) = \frac{(2\pi)^{-\left(\frac{n}{2}\right)}}{\sqrt{\text{Det}(\Sigma)}} \exp \left( -\frac{(\boldsymbol{x} - \boldsymbol{\mu})^\top \Sigma^{-1} (\boldsymbol{x} - \boldsymbol{\mu})}{2} \right),
\tag{22}
$$

where $\boldsymbol{x} \in \mathbb{R}^k$ is the variable vector, $\boldsymbol{\mu} \in \mathbb{R}^k$ is the mean vector, $\Sigma \in P_k(\mathbb{R})$ is the covariance matrix, and $P_k(\mathbb{R})$ is the space of $k$ positive definite symmetric matrices. We can define the statistical manifold by these distributions as $\mathcal{M} = \{p_{\boldsymbol{\theta}}; \boldsymbol{\theta} = (\boldsymbol{\mu}, \Sigma) \in \mathbb{R}^k \times P_k(\mathbb{R})\}$. By substituting equation (22) in equation (21), we can derive the Fisher information matrix for this parametrization, obtaining:

$$
g_{ij}(\boldsymbol{\theta}) = \frac{\partial \boldsymbol{\mu}^\top}{\partial \theta_i} \Sigma^{-1} \frac{\partial \boldsymbol{\mu}}{\partial \theta_j} + \frac{1}{2} \text{tr} \left( \Sigma^{-1} \frac{\partial \Sigma}{\partial \theta_i} \Sigma^{-1} \frac{\partial \Sigma}{\partial \theta_i} \right),
\tag{23}
$$

which induces the following square differential arc length in $\mathcal{M}$:

$$
ds^2 = d\boldsymbol{\mu}^\top \Sigma^{-1} d\boldsymbol{\mu} + \frac{1}{2} \text{tr} \left[ \left( \Sigma^{-1} d\Sigma \right)^2 \right].
\tag{24}
$$

Here, $d\boldsymbol{\mu} = (d\mu_1, \ldots, d\mu_n) \in \mathbb{R}^k$ and $d\Sigma = [d\sigma_{ij}] \in P_k(\mathbb{R})$. We observe that this metric is invariant to affine transformations (Pinele et al., 2020), i.e., for any $(\boldsymbol{c}, Q) \in \mathbb{R}^k \times GL_k(\mathbb{R})$, with $GL_k(\mathbb{R})$ the space of non-singular order $k$ matrices, the map $(\boldsymbol{\mu}, \Sigma) \mapsto (Q\boldsymbol{\mu} + \boldsymbol{c}, Q\Sigma Q^\top)$ is an isometry in $\mathcal{M}$. Thus, the Fisher-Rao distance between two multivariate normal distributions with parameters $\boldsymbol{\theta}_1 = (\boldsymbol{\mu}_1, \Sigma_1)$ and $\boldsymbol{\theta}_2 = (\boldsymbol{\mu}_2, \Sigma_2)$ in $\mathcal{M}$ satisfies:

$$d_{R,\mathcal{M}}(\boldsymbol{\theta}_1, \boldsymbol{\theta}_2) = d_{R,\mathcal{M}}\left((Q\boldsymbol{\mu}_1 + \boldsymbol{c}, Q\Sigma_1 Q^\top), (Q\boldsymbol{\mu}_2 + \boldsymbol{c}, Q\Sigma_2 Q^\top)\right). \tag{25}$$

Unfortunately, a closed-form solution for the Fisher-Rao distance remains unknown. This is still an open problem for an arbitrary covariance matrix $\Sigma$ and mean vector $\mu$. Fortunately, the FRD is known for the univariate case and hence, for the submanifold where $\Sigma$ is diagonal. Notice that in this case equation (24) admits an additive form.

From Pinele et al. (2020), we obtain the analytical expression of the Fisher-Rao in the 2-dimensional submanifold of univariate Gaussian probability distributions $\mathcal{M}_2 = \{p_{\boldsymbol{\theta}} : \boldsymbol{\theta} = (\mu, \sigma^2) \in \mathbb{R} \times (0, +\infty)\}$:

$$\rho_{\mathrm{FR}}\left((\mu_1, \sigma_1^2), (\mu_2, \sigma_2^2)\right) = \sqrt{2} \log \frac{\left|\left(\frac{\mu_1}{\sqrt{2}}, \sigma_1\right) - \left(\frac{\mu_2}{\sqrt{2}}, -\sigma_2\right)\right| + \left|\left(\frac{\mu_1}{\sqrt{2}}, \sigma_1\right) - \left(\frac{\mu_2}{\sqrt{2}}, \sigma_2\right)\right|}{\left|\left(\frac{\mu_1}{\sqrt{2}}, \sigma_1\right) - \left(\frac{\mu_2}{\sqrt{2}}, -\sigma_2\right)\right| - \left|\left(\frac{\mu_1}{\sqrt{2}}, \sigma_1\right) - \left(\frac{\mu_2}{\sqrt{2}}, \sigma_2\right)\right|}, \tag{26}$$

where $|\cdot|$ is the Euclidian norm in $\mathbb{R}^2$ and $\sigma$ denotes the standard deviation. Consequently, the FRD for Gaussian distributions with diagonal covariance matrix $\Sigma = \mathrm{diag}\left(\sigma_1^2, \sigma_2^2, \ldots, \sigma_k^2\right)$ in the $2k$-dimensional statistical submanifold $\mathcal{M}_D = \{p_{\boldsymbol{\theta}} : \boldsymbol{\theta} = (\boldsymbol{\mu}, \Sigma), \Sigma = \mathrm{diag}\left(\sigma_1^2, \sigma_2^2, \ldots, \sigma_k^2\right), \sigma_i > 0, i = 1, \ldots, k\}$ is

$$d_{\mathrm{FR-Gauss}}(\boldsymbol{\theta}_1, \boldsymbol{\theta}_2) = \sqrt{\sum_{i=1}^{k} d_{R,\mathcal{M}_2}\left((\mu_{1i}, \sigma_{1i}), (\mu_{2i}, \sigma_{2i})\right)^2}. \tag{27}$$

### A.3 Fisher-Rao vs. Mahalanobis distance

There is an intricate relationship between the FRD for multivariate Gaussian distributions and the Mahalanobis distance. We borrow the result from Pinele et al. (2020), which states that in the $k$-dimensional submanifold $\mathcal{M}_\Sigma$ of $\mathcal{M}$ where $\Sigma$ is constant, i.e., $\mathcal{M}_\Sigma = \{p_{\boldsymbol{\theta}} : \boldsymbol{\theta} = (\boldsymbol{\mu}, \Sigma), \Sigma = \Sigma_0 \in P_k(\mathbb{R})\}$, the Fisher-Rao distance $d_{R,\mathcal{M}_\Sigma}$ between two distributions is given by the Mahalanobis distance (Mahalanobis, 1936):

$$d_{R,\mathcal{M}_\Sigma}(\mathcal{N}(\boldsymbol{\mu}_1, \Sigma), \mathcal{N}(\boldsymbol{\mu}_2, \Sigma)) = \sqrt{(\boldsymbol{\mu}_1 - \boldsymbol{\mu}_2)^T \Sigma^{-1}(\boldsymbol{\mu}_1 - \boldsymbol{\mu}_2)}. \tag{28}$$

The Mahalanobis distance is also used for OOD detection (Lee et al., 2018) and its performance is compared to the FRD through several experiments in Section 4. Since the covariance matrix for the hidden layers' outputs is often not full rank, the pseudo-inverse is calculated instead of the inverse.

## B Igeood algorithms and computation details

In this section, we provide pseudo-code for calculating the IGEOOD score from the logits (Algorithm 1) and from the latent features (Algorithm 2). The BLACK-BOX IGEOOD score is obtained with Algorithm 1 by setting $\varepsilon = 0$, while the GREY-BOX IGEOOD score is obtained with $\varepsilon > 0$. We calculated the centroid of the logits for the in-distribution training set by optimizing the objective function given by equation (3) through a gradient descent algorithm for each DNN. We used a constant learning rate of $0.01$ and a batch size of $128$ for $100$ epochs. Finally, the WHITE-BOX IGEOOD score is obtained by combining the outputs of Algorithms 1 and 2 through fitting the multiplicative weights $\alpha$ through a logistic function classifier on a labeled mixture dataset composed from in- and out-of-distribution data according to a validation dataset, which leads to expression equation (13).

Note that the calculation of the training logits centroids $\boldsymbol{\mu}_y$, as well as the latent representations' mean vectors $\boldsymbol{\mu}_y^{(\ell)}$ and standard covariance matrices $\boldsymbol{\sigma}^{(\ell)}$ is performed beforehand, prior to inference. In this way, we retrieve the objects from memory at inference time. Also, we define $k$ as the cardinality of feature $\ell$, or $|f^{(\ell)}|$ and $\rho_{\mathrm{FR}}$ as the Fisher-Rao distance between univariate Gaussian distribution given by expression equation (9).

---

**Algorithm 1:** Evaluating IGEOOD score based on the logits.

---

**Input** : Test sample $\boldsymbol{x}$, temperature $T$ and noise magnitude $\varepsilon$ parameters, and training set $\mathcal{D}_N = \{(\boldsymbol{x}_i, y_i)\}_{i=1}^N$.

**Output:** $\mathrm{FR}_0$: IGEOOD score in the logits level.

// Offline computation

Calculate the logits centroids from the training data:

$\boldsymbol{\mu}_y \triangleq \min_{\boldsymbol{\mu} \in \mathbb{R}^C} \frac{1}{N_y} \sum_{\forall\, i\,:\, y_i = y} 2 \arccos \left( \sum_{y' \in \mathcal{Y}} \sqrt{q_{\boldsymbol{\theta}}\left(y'|f(\boldsymbol{x}_i)\right) q_{\boldsymbol{\theta}}\left(y'|\boldsymbol{\mu}\right)} \right)$

// Online computation

Add small perturbation to $\boldsymbol{x}$:

$\widetilde{\boldsymbol{x}} \leftarrow \boldsymbol{x} + \varepsilon \odot \mathrm{sign}\left[ \nabla_{\boldsymbol{x}} \sum_y 2 \arccos \left( \sum_{y' \in \mathcal{Y}} \sqrt{q_{\boldsymbol{\theta}}(y'|f(\boldsymbol{x})) q_{\boldsymbol{\theta}}(y'|\boldsymbol{\mu}_y)} \right) \right]$

**return** $\mathrm{FR}_0(\widetilde{\boldsymbol{x}}) \leftarrow \sum_y 2 \arccos \left( \sum_{y' \in \mathcal{Y}} \sqrt{q_{\boldsymbol{\theta}}(y'|f(\widetilde{\boldsymbol{x}})) q_{\boldsymbol{\theta}}(y'|\boldsymbol{\mu}_y)} \right)$

---

**Algorithm 2:** Evaluating feature-wise IGEOOD score.

---

**Input** : Test sample $\boldsymbol{x}$ and training set $\mathcal{D}_N = \{(\boldsymbol{x}_i, y_i)\}_{i=1}^N$.

**Output:** $\mathrm{FR}_\ell$: feature-wise IGEOOD scores.

**for** *each feature $\ell \in \{1, \dots, L\}$* **do**

    // Offline computation

    Calculate the means: $\boldsymbol{\mu}_y^{(\ell)} \leftarrow \frac{1}{N_y} \sum_{i:y_i=y} f^{(\ell)}\left(\boldsymbol{x}_i\right)$

    Calculate the diagonal standard deviation matrix:

    $\sigma_{jj}^{(\ell)} \leftarrow \sqrt{\frac{1}{N} \sum_{y \in \mathcal{Y}} \sum_{\forall i\,:\, y_i = y} \left( f_j^{(\ell)}\left(\boldsymbol{x}_i\right) - \mu_{y,j}^{(\ell)} \right)^2}$

    // Online computation

    Compute the OOD score for $\ell$:

    $\mathrm{FR}_\ell(\boldsymbol{x}) \leftarrow \min_y \sqrt{\sum_{j=1}^k \rho_{\mathrm{FR}}\left( \left( \mu_{y,j}^{(\ell)}, \sigma_{jj}^{(\ell)} \right), \left( f_j^{(\ell)}(\boldsymbol{x}), \sigma_{jj}^{(\ell)} \right) \right)^2}$

**end**

**return** $\left( \mathrm{FR}_1(\boldsymbol{x}), \dots, \mathrm{FR}_L(\boldsymbol{x}) \right)$

---

### B.1 LOGITS CENTROIDS ESTIMATION DETAILS

In order to obtain the logits centroids given the Fisher-Rao distance in the space of softmax probability distributions, we designed a simple optimization problem. This problem aims to minimize the average distance between the class conditional training samples and the centroids as given by equation (3). We initialized the $C$ centroids, where $C$ is the number of classes of a given model, with the identity matrix of size $C \times C$. Note that the initial centroid for class $i$ is given by the matrix's line number $i$. We minimized the expression in equation (3) with a gradient descent optimizer for 100 epochs with a fixed learning rate equal to 0.1 for every DNN model and in-distribution dataset.

The computation of the logits centroid is done offline, and the loss of the centroid estimation converges fast. We show in Table 3 the execution time for some operations in the OOD detection pipeline accelerated by one GPU. The left-hand column shows the offline computations needed to run our setup. They are as follow:

- Save train set logits: We first do a forward pass through all the training sets and save in memory the resulting logits for a given network, which takes on average 83s for CIFAR-10 and CIFAR-100;

- Centroid estimation: We load the training logits from memory and run the Gradient Descent algorithm, which takes on average 1.2s for CIFAR-10 and 11s for CIFAR-100;

---

**Algorithm 3:** Evaluating feature-wise IGEOOD+ score.

---

**Input** : Test sample $\boldsymbol{x}$, training set $\mathcal{D}_N = \{(\boldsymbol{x}_i, y_i)\}_{i=1}^N$ and $M$ OOD samples
$\mathcal{O}_M = \{\boldsymbol{x}_i'\}_{i=1}^M$.

**Output:** $\mathrm{FR}_\ell$ and $\mathrm{FR}_\ell'$: feature-wise IGEOOD+ scores.

---

**for** *each feature $\ell \in \{1, \ldots, L\}$* **do**

$\quad$ // Offline computation

$\quad$ Calculate class conditional means: $\boldsymbol{\mu}_y^{(\ell)} \leftarrow \frac{1}{N_y} \sum_{i:y_i=y} f^{(\ell)}(\boldsymbol{x}_i)$

$\quad$ Calculate OOD samples mean: $\boldsymbol{\mu}^{(\ell)\prime} \leftarrow \frac{1}{M} \sum_{i=1}^M f^{(\ell)}(\boldsymbol{x}_i')$

$\quad$ Calculate the diagonal standard deviation matrix from training data:

$\quad$ $\sigma_{jj}^{(\ell)} \leftarrow \sqrt{\frac{1}{N} \sum_{y \in \mathcal{Y}} \sum_{\forall i\,:\,y_i=y} \left( f_j^{(\ell)}(\boldsymbol{x}_i) - \mu_{y,j}^{(\ell)} \right)^2}$

$\quad$ Calculate the diagonal standard deviation matrix from OOD data:

$\quad$ $\sigma_{jj}^{(\ell)\prime} \leftarrow \sqrt{\frac{1}{M} \sum_{i=i}^M \left( f_j^{(\ell)}(\boldsymbol{x}_i') - \mu_j^{(\ell)\prime} \right)^2}$

$\quad$ // Online computation

$\quad$ Compute the OOD scores for $\ell$:

$\quad$ $\mathrm{FR}_\ell(\boldsymbol{x}) \leftarrow \min_y \sqrt{\sum_{j=1}^k \rho_{\mathrm{FR}} \left( \left( \mu_{y,j}^{(\ell)}, \sigma_{jj}^{(\ell)} \right), \left( f_j^{(\ell)}(\boldsymbol{x}), \sigma_{jj}^{(\ell)} \right) \right)^2}$

$\quad$ $\mathrm{FR}_\ell'(\boldsymbol{x}) \leftarrow \min_y \sqrt{\sum_{j=1}^k \rho_{\mathrm{FR}} \left( \left( \mu_j^{(\ell)\prime}, \sigma_{jj}^{(\ell)\prime} \right), \left( f_j^{(\ell)}(\boldsymbol{x}), \sigma_{jj}^{(\ell)} \right) \right)^2}$

**end**

**return** $\left( \mathrm{FR}_1(\boldsymbol{x}), \mathrm{FR}_1'(\boldsymbol{x}) \ldots, \mathrm{FR}_L(\boldsymbol{x}), \mathrm{FR}_L'(\boldsymbol{x}) \right)$

---

The right-hand side of Table 3 shows the average online computation time for one test sample in a BLACK-BOX setting.

- Model inference: The average time needed to complete one forward pass for a DenseNet-BC-100 model is 28 ms and 19 ms for CIFAR-10 and CIFAR-100, respectively.
- MSP and BLACK-BOX IGEOOD computations. Computing the OOD detection scores from the calculated softmax output is roughly 100 to 1000 times faster than the inference time taken by the model.

Hence, computing the Fisher-Rao distance between a test sample and the class-conditional centroids does not account for a considerable overhead in execution time.

Table 3: Execution time analysis for an experimental set accelerated by a single GPU for a DenseNet-BC-100 architecture pre-trained on CIFAR-10 and CIFAR-100. We show the average value for 5 runs.

| In-dist. Dataset | Offline computation | | Online computation | | |
| --- | --- | --- | --- | --- | --- |
| | Save train set logits | Centroid estimation | Model inference | MSP computation | BLACK-BOX IGEOOD computation |
| CIFAR-10 | 83 s | 1.2 s | 28 ms | 63 $\mu$s | 66 $\mu$s |
| CIFAR-100 | 83 s | 11 s | 19 ms | 34 $\mu$s | 171 $\mu$s |

## B.2 COVARIANCE MATRIX ESTIMATION DETAILS

We model the latent output probability distributions as Gaussian distributions with diagonal covariance matrix calculated with equation (8). We chose this model motivated by a closed form for

the FRD and by observing that the standard covariance matrix for the latent features is often ill-conditioned and diagonal dominant. The condition number of a matrix correlates to its numerical stability, i.e., a small rounding error in its estimation may cause a large difference in its values. So, a matrix with a low condition number is said to be well-conditioned, while a matrix with a high condition number is said to be ill-conditioned. We calculate the condition number of the covariance matrices with the formula $\kappa(\Sigma) = \left\|\Sigma^{-1}\right\|_{\infty} \|\Sigma\|_{\infty}$, where $\|\cdot\|_{\infty}$ is the infinity norm. For each of the four dense blocks outputs of a DenseNet trained on CIFAR-10, we obtained the condition numbers $\kappa_{\Sigma} = \{2.8\mathrm{e}10, 3.5\mathrm{e}6, 3.1\mathrm{e}5, 3.5\mathrm{e}21\}$. While for the *diagonal* covariance matrix, we obtained smaller values of condition numbers: $\kappa_{\Sigma_D} = \{1.0\mathrm{e}3, 3.0\mathrm{e}1, 1.4\mathrm{e}1, 7.6\mathrm{e}20\}$. We associate the high value for the last feature mainly because the last feature is high dimensional and coarse, i.e., most of the values in the diagonal are close to zero.

### B.3 Gaussianity test of the hidden layers' outputs

In order to test if the Gaussian assumption is valid for the outputs of the hidden feature, we conduct a Shapiro & Wilk (1965) normality test for each coordinate of the features for the training data of a DenseNet model. We calculated the test's $W$ statistic for each coordinate and class and averaged them. We chose a univariate normality test because they are often powerful and the problem is high dimensional, which would be unfavorable for a multivariate statistic test. Thus, this study should be considered with caution, given the considered hypothesis. In Figure 3, we also show the standardized histograms for the first coordinate of each layer. Note that, apart from the penultimate layer, if we consider the coordinates of the hidden features independently, the Gaussianity assumption holds, as we obtain a $W$ statistics close to 1. However, for the last block, this assumption sometimes does not hold. Hence, modeling the penultimate layer with a more powerful density estimator, and using a metric that considers this more complex distribution, may be favorable for OOD detection.

### B.4 Feature importance regression details

For both Mahalanobis and IGEOOD methods, we fitted a logistic regression model with cross-validation using 1,000 OOD and 1,000 in-distribution data samples. Each regression parameter multiplies the layer scores outputs with the objective function of maximizing the TNR at TPR-95%. We set the maximum number of iterations to 100.

In order to investigate which hidden feature assists the most in OOD detection, we calculate the TNR at TPR-95% for the scores in the outputs of Blocks 1, 2 and 3 of a DenseNet pre-trained on CIFAR-10. We took as OOD data the SVHN dataset. Figure 4 shows the histogram and detection performance for each layer as well as the results from the logistic regression. Note that for the IGEOOD score in this study, we did not consider the logits.

## C Detailed Experimental Setup

### C.1 DNN models and training details

We describe the DNN models used in the experiments:

- **DenseNet.** Densely Connected Convolutional Networks (Huang et al., 2017), or DenseNet for short, are compositions of dense blocks, which are composed of multiple layers directly connected to every other layer in a feed-forward fashion. In this work, we use the DenseNet-BC-100 architecture. The BC stands for a model with 1x1 convolutional bottleneck (B) layers and channel number compression (C) of 0.5. The models have depth $L = 100$ and growth rate $k = 12$. We consider the outputs of each dense block after the transition layer (3 in total) and the first convolutional layer output as the latent features. After an averaging pooling, the latent features have dimensions $\mathcal{F}_1 = \{24, 108, 150, 342\}$.

- **ResNet.** Residual Networks (He et al., 2016), or ResNet, are deep neural networks composed of residual blocks. Each residual block is composed of layers connected in a feed-forward manner plus a skip connection. We use the ResNet with 34 layers pre-trained on CIFAR-10, CIFAR-100, and SVHN datasets. We take the output of every residual block (4 in total) and the first convolutional layer for calculating the score on

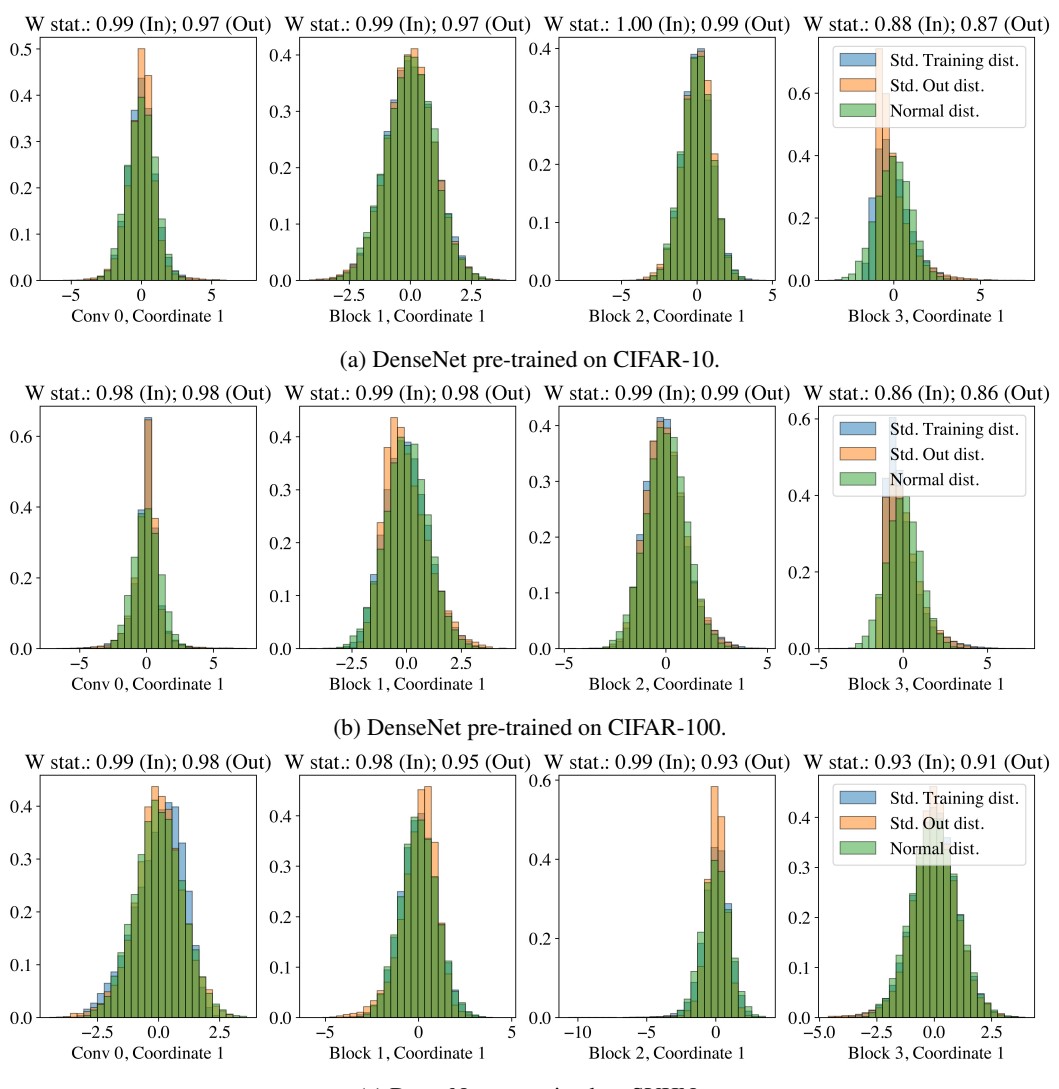

Figure 3: Histograms of the standardized first coordinate output of each hidden feature of a DenseNet model for in-distribution and out-of-distribution (TinyImageNet) compared to a 1-D Normal distribution. The Average Shapiro-Wilk test's W statistics is close to one for Conv 0, Block 1 and Block 2, which indicates that the coordinates, and potentially the feature vector, are provably Gaussian. The penultimate layer (outputs of Block 3) has a lower test statistic for the given experiments.

the WHITE-BOX setting. After an averaging pooling, the latent features have dimensions $\mathcal{F}_2 = \{64, 64, 128, 256, 512\}$.

We train each model by minimizing the cross-entropy loss using SGD with Nesterov momentum equal to 0.9, weight decay equal to 0.0001, and a multi-step learning rate schedule starting at 0.1 for 300 epochs. The pre-trained models is available at [2]. We report their test set accuracy in Table 4 with the softmax function and by replacing it with the Fisher-Rao distance between the training class-conditional centroids and the test sample outputs. Also, it is worth noting that one high-end GPU is sufficient for running every experiment presented in this work.

---

[2] https://github.com/edadaltocg/Igeood

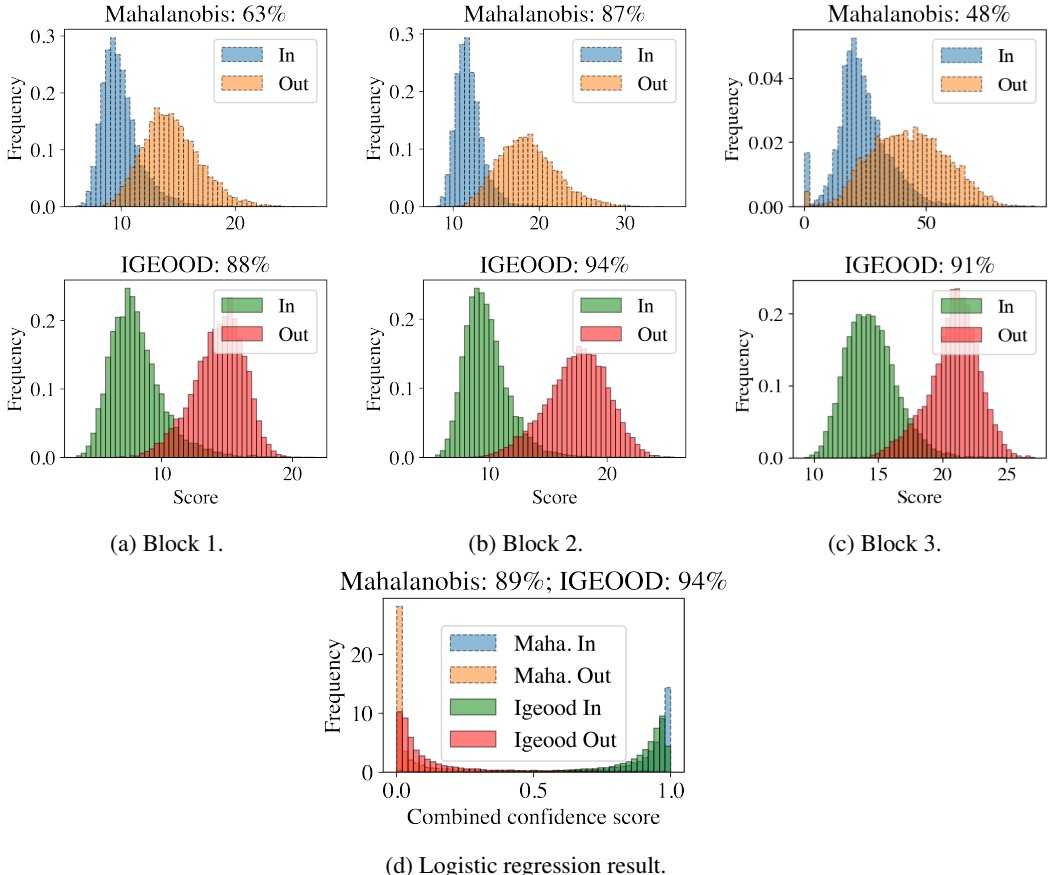

(a) Block 1.  (b) Block 2.  (c) Block 3.

(d) Logistic regression result.

Figure 4: Histograms of the Mahalanobis and IGEOOD scores for the output of each hidden block of a DenseNet model for CIFAR-10 (in-dstribution) and SVHN (out-of-distribution). The title shows the TNR at TPR-95% considering only the scores of the outputs of the given layer. The logistic regression found as coefficients: $\boldsymbol{\alpha} = (1.0, -3.6, -0.13)$ for Mahalanobis and $\boldsymbol{\alpha} = (1.0, 1.3, 1.2)$ for IGEOOD.

Table 4: Test set accuracy in percentage for ResNet and DenseNet architectures pre-trained on CIFAR-10, CIFAR-100 and SVHN.

| In-Dataset | ResNet-34 | | DenseNet-BC-100 | |
|---|---|---|---|---|
| | Softmax | Fisher-Rao | Softmax | Fisher-Rao |
| CIFAR-10 | 93.52 | **93.53** | 95.20 | 95.20 |
| CIFAR-100 | **77.11** | 77.09 | 77.62 | **77.63** |
| SVHN | 96.61 | 96.61 | 95.16 | 95.16 |

## C.2 EVALUATION METRICS

We introduce below standard binary classification performance metrics used to evaluate the OOD discriminators.

- **True Negative Rate at 95% True Positive Rate (TNR at TPR-95% (%)).** This metric measures the true negative rate (TNR) at a specific true positive rate (TPR). The operating point is chosen such that the TPR of the in-distribution test set is fixed to some value, 95% in this case. Mathematically, let TP, TN, FP, and FN denote true positive, true negative, false positive and false negative, respectively. We measure TNR = TN/(FP + TN), when TPR = TP/(TP + FN) is 95%.

- **Area Under the Receiver Operating Characteristic curve (AUROC (%)).** The ROC curve is constructed by plotting the true positive rate (TPR) against the false positive rate ($= FP/(FP + TN)$) at various threshold values. The area under this curve tells how much the OOD discriminator can distinguish in-distribution and OOD data in a threshold-independent manner.

- **Area Under the Precision-Recall curve (AUPR (%)).** The PR curve plots the precision ($= TP/(TP + FP)$) against the recall ($= TP/(TP + FN)$) by varying a threshold. For the experiments, in-distribution data are specified as positives while OOD data as negative.

Note that the TNR at TPR-95% is significant because we want to identify OOD data and preserve a sufficiently good performance on identifying in-distribution data, which is not the case for the other metrics.

### C.3 DATASETS

We use natural image examples from the following image classification and synthetic datasets in our experiments. We normalize the test samples with the in-distribution dataset statistics.

- **CIFAR-10.** The CIFAR-10 (Krizhevsky et al., 2009) dataset is composed of $32 \times 32$ natural images of 10 different classes, e.g., airplane, ship, bird, etc. The training set comprises 50,000 images, and the test set is composed of 10,000 images. The classes are approximately equally distributed (5,000 examples each label). The CIFAR-10 dataset is under the MIT license.

- **CIFAR-100.** The CIFAR-100 (Krizhevsky et al., 2009) dataset contains similar natural images to the CIFAR-10 dataset, but with 90 additional categories. Its set repartition is 50,000 for training and 10,000 for the test set. We expect around 500 samples for each class of the training set. It is also under the MIT license.

- **SVHN.** The SVHN (Netzer et al., 2011) dataset collects street house numbers for digit classification. It contains 73,257 training and 26,032 test RGB images of size $32 \times 32$ of printed digits (from 0 to 9). We take only the first 10,000 examples of the test set for evaluating the methods to have a balanced dataset of in-distribution and out-of-distribution data. This dataset is subject to a non-commercial license.

- **Tiny-ImageNet.** The Tiny-ImageNet (Le & Yang, 2015) dataset is a subset of the large-scale natural image dataset ImageNet (Deng et al., 2009). It contains 200 different classes and 10,000 test examples. We downsize the images from their original resolution to images of dimension $32 \times 32 \times 3$.

- **LSUN.** The LSUN (Yu et al., 2015) dataset, which has equally 10,000 test examples, is used for the large-scale scene classification of different scene categories (e.g., bedroom, bridge, kitchen, etc.). Similarly, we resize the images following the same procedure for the Tiny-ImageNet dataset. LSUN is under the Apache 2.0 license.

- **iSUN.** The iSUN (Xu et al., 2015) dataset consists of selected natural scene images from the SUN (Xiao et al., 2010) dataset. The test set has 8925 images, which we downsample to $32 \times 32 \times 3$. We use this dataset as a source of OOD for validation purposes as an independent dataset from the test OOD data.

- **Textures.** The Describable Textures Dataset (DTD) (Cimpoi et al., 2014) is a collection of textural pattern images observed in nature. It contains 47 categories totaling 5640 images of various sizes, which are resized and center cropped to fit into the input size of $32 \times 32$.

- **Chars74K.** The Chars74K dataset (de Campos et al., 2009) contains 74,000 samples of 62 classes of characters found in natural images, handwritten text, and synthesized from computer fonts. We used as OOD data only the *EnglishImg* dataset split, which contains 7705 characters from natural scenes. We resized and center-cropped the images.

- **Places365.** The Places365 dataset (Zhou et al., 2017) contains images of 365 natural scenes categories. We used the small images validation split as OOD data in our experiments. It contains 36,500 RGB images which were downsampled from $256 \times 256$ to $32 \times 32$.

- **Gaussian.** For the Gaussian dataset, we generated 10,000 synthetic RGB images from 2D Gaussian noise, where each RGB pixel is sampled from an i.i.d Gaussian distribution with mean 0.5 and variance 1.0. The pixel values are clipped to $[0, 1]$ interval. This synthetic data was introduced in previous work as an easy benchmark (Hendrycks & Gimpel, 2017).

### C.4 ADVERSARIAL DATA GENERATION

We generate adversarial samples from the in-distribution dataset using the fast gradient sign method (FGSM). This method works by exploiting the gradients of the neural network to create a non-targeted adversarial attack. For an input image $x_i$, the method computes the sign of the gradients of the loss function $J$ with respect to the input image to create a new image $x_i^{\mathrm{adv}}$ that maximizes the loss as given by equation (29). This fabricated image is called an adversarial image, which we use for tuning the hyperparameters of the OOD detection methods in the WHITE-BOX case. Mathematically,

$$x_i^{\mathrm{adv}} = x_i + \varepsilon^{\mathrm{adv}} \odot \mathrm{sign}(\nabla_{x_i} J(\theta, x_i, y_i)), \qquad (29)$$

where $\varepsilon^{\mathrm{adv}} > 0$ is the additive noise magnitude parameter. Table 5 shows the resulting $L_\infty$ mean perturbation and classification accuracy on adversarial samples.

Table 5: The $L_\infty$ mean perturbation used to generate adversarial data with FGSM algorithm and classification accuracy on adversarial samples for the DNN models and in-distribution datasets.

|                | CIFAR-10 | | CIFAR-100 | | SVHN | |
|----------------|----------|------|-----------|--------|------|------|
|                | $L_\infty$ | Acc. | $L_\infty$ | Acc. | $L_\infty$ | Acc. |
| DenseNet-BC-100 | 0.21 | 19.5% | 0.20 | 4.45% | 0.32 | 54.7% |
| ResNet-34      | 0.21 | 23.7% | 0.20 | 12.49% | 0.25 | 50.0% |

## D   BENCHMARK METHODS

This section briefly introduces the benchmark OOD detection methods with a standardized notation.

### D.1   BASELINE

DNNs tend to assign lower confidence for OOD samples. So, calculating the Maximum Softmax Probability (MSP) (Hendrycks & Gimpel, 2017) is a natural baseline for OOD detection. In other words, provided an input data $x$, a pre-trained neural network $f(\cdot)$, and a confidence threshold $\delta$, the OOD score, and the discriminator are given by

$$s(x) = \max_{y \in \mathcal{Y}} \frac{e^{f_y(x)}}{\sum_{y' \in \mathcal{Y}} e^{f_{y'}(x)}} \quad \text{and} \quad S(x; \delta) = \begin{cases} 1 & \text{if } s(x) \leq \delta \\ 0 & \text{if } s(x) > \delta \end{cases}, \qquad (30)$$

respectively. Here, $f_y(x)$ indicates the $y$-th logits output. A limitation of this method is that unscaled softmax posterior distributions are usually spiky, i.e., softmax trained deep neural models are incorrectly calibrated, which does not favor OOD detection (Lakshminarayanan et al., 2017).

### D.2   ODIN: OOD DETECTOR FOR NEURAL NETWORKS

In summary, ODIN (Liang et al., 2018) explores the weaknesses of the MSP criterion by recalibrating the output's confidence to the task of OOD detection. They improve the MSP baseline by using the temperature scaled softmax function (equation (1)) instead. Also, ODIN adds small adversarial noise perturbation to the inputs, i.e.,

$$\widetilde{x} = x - \varepsilon \odot \mathrm{sign}\left(-\nabla_x \log q_\theta(y|f(x); T)\right), \qquad (31)$$

where $\varepsilon$ is the perturbation magnitude. Hyperparameters $T$ and $\varepsilon$ are tuned on a validation dataset without requiring prior knowledge of test OOD data. They calculate the confidence score by taking the maximum of the perturbed input temperature scaled softmax outputs.

### D.3 Energy-based OOD detector

An energy-based OOD discriminator is proposed by Liu et al. (2020), where the differences of energies between in-distribution and OOD samples allow for distribution distinction. The energy-based model substitutes the softmax function with the Helmholtz free energy equation to extract a confidence score. They observed that examples with higher energy have a low likelihood of occurrence, concluding that they are likely OOD. The free energy expression is:

$$E(\boldsymbol{x}; f) = -T \cdot \log \sum_{y \in \mathcal{Y}} e^{f_y(\boldsymbol{x})/T}. \tag{32}$$

Note that, differently from ODIN and MSP, they use the information of all of the logits output values through the sum operation. Besides, they apply input pre-processing for further separating OOD data from in-distribution.

### D.4 Mahalanobis distance-based confidence score

The Mahalanobis-based method in Lee et al. (2018) fits the DNN training data features as class-conditional Gaussian distributions. These use the outputs of every DNN latent block to leverage useful information for discrimination. For a test sample $\boldsymbol{x}$, the confidence score from the $\ell$-th feature is calculated based on the Mahalanobis distance between $f^{(\ell)}(\boldsymbol{x})$ and the closest class-conditional distribution:

$$M_\ell(\boldsymbol{x}) = \max_y - \left(f^{(\ell)}(\boldsymbol{x}) - \widehat{\boldsymbol{\mu}}_y^{(\ell)}\right)^\top \widehat{\Sigma}_\ell^{-1} \left(f^{(\ell)}(\boldsymbol{x}) - \widehat{\boldsymbol{\mu}}_y^{(\ell)}\right), \tag{33}$$

where $f^{(\ell)}(\boldsymbol{x})$ is the $\ell$-th latent feature output, and $\widehat{\boldsymbol{\mu}}_y^{(\ell)}$ and $\widehat{\Sigma}_\ell$ are, respectively, the empirical class mean and covariance matrix estimates. The covariance matrix is often not full rank, so the pseudo-inverse is calculated instead of the inverse. In addition, input pre-processing and feature ensemble are also used to boost performance. A logistic regression model learns the multiplicative weights $\alpha_\ell$ for each layer score, which predicts 1 for in-distribution and 0 for OOD examples from a mixture validation dataset. Finally, the Mahalanobis-based discriminator is given by thresholding expression $\sum_\ell \alpha_\ell M_\ell(\boldsymbol{x})$.

## E Additional Out-Of-Distribution detection results

### E.1 Fisher-Rao distance versus Kullback-Leibler Divergence

From Picot et al. (2021), the Kullback-Leibler divergence (KL) is connected to the Fisher-Rao distance between softmax probability distributions ($d_{R,\mathcal{D}}$) by the inequality:

$$1 - \cos\left(\frac{d_{R,\mathcal{D}}\left(q_{\boldsymbol{\theta}}, q'_{\boldsymbol{\theta}}\right)}{2}\right) \leq \frac{1}{2} \mathrm{KL}\left(q_{\boldsymbol{\theta}}, q'_{\boldsymbol{\theta}}\right). \tag{34}$$

To verify how the KL divergence would behave for OOD detection, we ran experiments with our Black-Box setting, where we calculated the class conditional centroids with the KL divergence. We calculated the divergence of the test sample w.r.t each of these centroids during test time, then aggregated the results with a sum or by taking the minimal value. The results are displayed in Table 6. We can conclude from these experiments that taking the sum of the outputs instead of the minimal value is overall advantageous for Fisher-Rao distance and KL divergence.

### E.2 Hyperparameters tuning

For temperature $T$, we ran a Bayesian optimization for 500 epochs in the interval of temperature values between 1 and 1000, where the objective function was to maximize the TNR at TPR-95% metric for the validation set. We took the best temperature among five runs with different random seeds. For the input pre-processing noise magnitude $\varepsilon$ tuning, we ran a grid search optimization with 21 equally spaced values in the interval $[0, 0.002]$. Table 7 shows the best hyperparameters we found for the methods in the Black-Box, Grey-Box, and White-Box settings.

Table 6: Performance comparison between the Fisher-Rao distance and the KL Divergence for OOD detection in a BLACK-BOX setting. The numerical values in the Table are TNR at TPR-95% in percentage for a DenseNet and ResNet models pre-trained on CIFAR-10, CIFAR-100 and SVHN datasets. FISHER-RAO (sum) corresponds to the IGEOOD score.

| | OOD dataset | CIFAR-10 | CIFAR-100 | SVHN |
|---|---|---|---|---|
| | | FISHER-RAO (sum) / FISHER-RAO (min) / KL (min) / KL (sum) | | |
| DenseNet | Chars | 55.1/45.0/**56.9**/54.6 | 17.2/14.6/20.1/17.1 | 47.9/46.6/**50.1**/46.9 |
| | Gaussian | **99.9**/97.9/97.9/**99.9** | 0.0/0.0/0.0/0.0 | 98.0/97.2/73.1/**98.1** |
| | TinyImgNet | 87.8/73.6/72.3/**88.1** | **25.7**/18.1/15.7/25.4 | **85.1**/84.1/69.4/85.0 |
| | LSUN | 93.3/81.9/86.4/**93.4** | 25.4/17.8/15.1/25.2 | 85.0/83.9/66.2/85.4 |
| | Places365 | 52.2/49.7/**57.2**/51.5 | 20.7/**20.9**/18.2/20.8 | **71.9**/71.1/59.1/71.3 |
| | Textures | 35.8/46.9/**51.0**/34.8 | 22.8/19.5/17.6/**23.1** | 56.5/**57.4**/65.3/55.4 |
| | CIFAR-10 | - | 17.0/**20.0**/18.5/17.7 | **67.0**/66.0/56.2/65.8 |
| | CIFAR-100 | **50.8**/48.7/49.6/50.3 | - | 65.4/**65.0**/59.1/64.1 |
| | SVHN | 50.1/47.9/**50.7**/49.6 | **36.7**/29.9/29.1/35.9 | - |
| | average | **65.6**/61.4/65.3/65.3 | **20.7**/17.6/16.8/20.6 | **72.1**/71.4/62.3/71.5 |
| ResNet | Chars | **51.1**/45.0/41.6/49.6 | 15.2/14.6/14.5/**15.5** | 58.5/57.4/46.2/**58.4** |
| | Gaussian | **89.0**/86.4/62.3/86.8 | 0.6/1.7/**3.9**/0.9 | 87.3/**87.5**/76.7/87.0 |
| | TinyImageNet | **58.2**/51.4/51.4/57.8 | **23.0**/17.8/10.4/21.6 | **82.2**/81.6/68.8/81.9 |
| | LSUN | **62.0**/53.8/56.0/**62.0** | **20.6**/15.4/10.2/19.5 | 77.4/**77.5**/64.6/77.2 |
| | Places365 | **48.2**/40.0/39.6/48.1 | 16.9/17.3/16.7/**17.8** | 79.0/**79.1**/67.2/78.8 |
| | Textures | **50.3**/44.0/45.6/49.8 | **23.4**/20.9/14.1/23.2 | 80.9/**80.9**/72.8/80.6 |
| | CIFAR-10 | - | 18.0/**18.1**/16.8/18.8 | **81.2**/81.0/67.8/81.1 |
| | CIFAR-100 | **45.9**/38.9/36.8/45.6 | - | **80.2**/79.8/66.2/79.9 |
| | SVHN | **48.8**/31.6/31.5/47.0 | 13.3/14.3/**15.7**/14.3 | - |
| | average | **56.7**/48.9/45.6/55.8 | **16.4**/15.0/12.8/**16.4** | **78.3**/78.1/66.3/78.1 |

Table 7: Best temperatures $T$ for the BLACK-BOX setup, best temperature and noise magnitude $(T, \varepsilon)$ for the GREY-BOX setup, and best $\varepsilon$ for the Mahalanobis score and $(T, \varepsilon)$ for IGEOOD and IGEOOD+ in the WHITE-BOX setup with adversarial tuning.

| Model | In-dist. dataset | BLACK-BOX | | | GREY-BOX | | | WHITE-BOX | |
|---|---|---|---|---|---|---|---|---|---|
| | | ODIN | Energy | IGEOOD | ODIN | Energy | IGEOOD | Maha. | IGEOOD,+ |
| DenseNet | C-10 | 1000 | 4.6 | 5.3 | (1000, 0.0014) | (4.6, 0.0012) | (5.3, 0.0012) | 0 | (5, 0.0015) |
| | C-100 | 1000 | 1.1 | 2.1 | (1000, 0.0020) | (1.1, 0.0020) | (2.1, 0.0020) | 0 | (5, 0) |
| | SVHN | 1 | 1.1 | 1.1 | (1, 0.0010) | (1.1, 0.0006) | (1.1, 0.0006) | 0.001 | (5, 0.0015) |
| ResNet | C-10 | 1000 | 5.4 | 5.3 | (1000, 0.0014) | (5.4, 0.0012) | (5.3, 0.0012) | 0.0005 | (2, 0) |
| | C-100 | 1000 | 1 | 1 | (1000, 0.0020) | (9.1, 0.0024) | (12.7, 0.0024) | 0.0005 | (1, 0) |
| | SVHN | 1000 | 1.7 | 1 | (1000, 0.0004) | (1.7, 0.0002) | (1.0, 0.0004) | 0 | (5, 0) |

E.3 TEMPERATURE SCALING AND NOISE MAGNITUDE PLOTS

In Figure 5 and 6, we plot on the left hand side column the effect of the temperature parameter in the performance for the BLACK-BOX setup. We set the noise magnitude to zero and measured the TNR at TPR-95% for 500 different temperatures values found by a Bayesian optimization for a variety of DNN models. The performance is evaluated on the iSUN dataset. The right hand side column of Figure 5 and 6 show the effect of the noise magnitude parameter in the performance of IGEOOD score in the GREY-BOX setup. We set the temperature to the best found in the BLACK-BOX case. Then, we measured the OOD performance for 21 values of noise magnitude $\varepsilon$ equally spaced in the interval $[0, 0.004]$. The best couple $(T, \varepsilon)$ for each method and model is used to evaluate the GREY-BOX performances. The best hyperparameters found are detailed in Table 7.

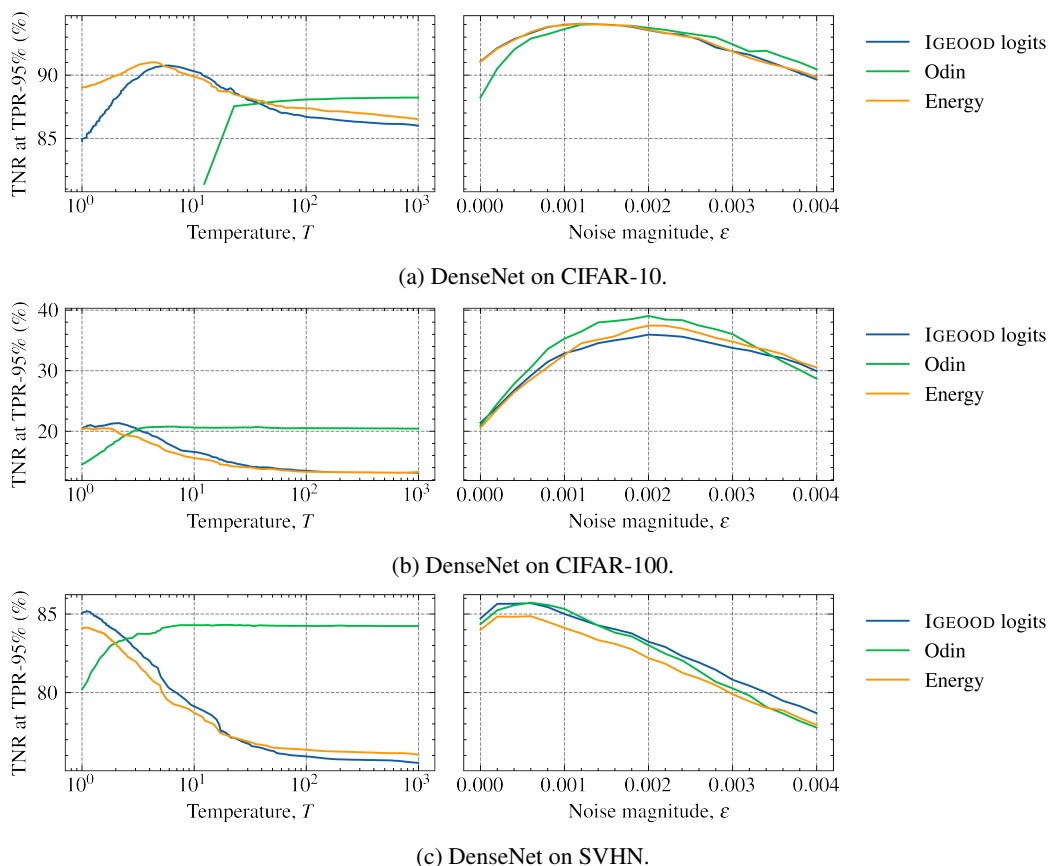

(a) DenseNet on CIFAR-10.

(b) DenseNet on CIFAR-100.

(c) DenseNet on SVHN.

Figure 5: OOD detection performance against temperature and noise magnitude parameters for ODIN (Liang et al., 2018), Energy (Liu et al., 2020) and IGEOOD (ours) on the iSUN (Xu et al., 2015) OOD dataset for a DenseNet-100 architecture.

### E.4 CONSISTENCY OF IGEOOD SCORE CONCERNING THE CHOICE OF THE VALIDATION DATA

To verify the consistency of IGEOOD and other methods to the choice of validation data, we measured the TNR at TPR-95% after tuning our method in a BLACK-BOX and GREY-BOX scenario on nine validation datasets. In Table 8, the first column shows the validation dataset, while we used the remaining OOD datasets to evaluate performance. We obtained consistent results, ranging from 63.4% to 72.0% the average TNR at TPR-95% in the BLACK-BOX case and from 65.0% to 73.4% in the GREY-BOX setting. We show that input pre-processing provides mild amelioration for our method and can be considered a fine-tuning step.

### E.5 ERROR BARS AND STANDARD DEVIATION

We conduct all of our experiments during inference time. Provided that we fix the DNN, the in-distribution, and the out-of-distribution datasets, there is not a source of randomness to our algorithm because the weights $\alpha$ of the feature ensemble method and centroids are initialized deterministically. Thus, the OOD scores for the same experimental setting do not change. To confirm this, we ran the same experiment five times and obtained the same results in all of them. However, if we allow for retraining the DNN from scratch, we might obtain different parameters, leading to slightly different model accuracy and potentially OOD detection performance. With this in mind, we retrained a DenseNet-BC-100 model on CIFAR-10 five times with five different random seeds. The results for OOD detection in a BLACK-BOX setting for the 5 models can be found in Table 9.

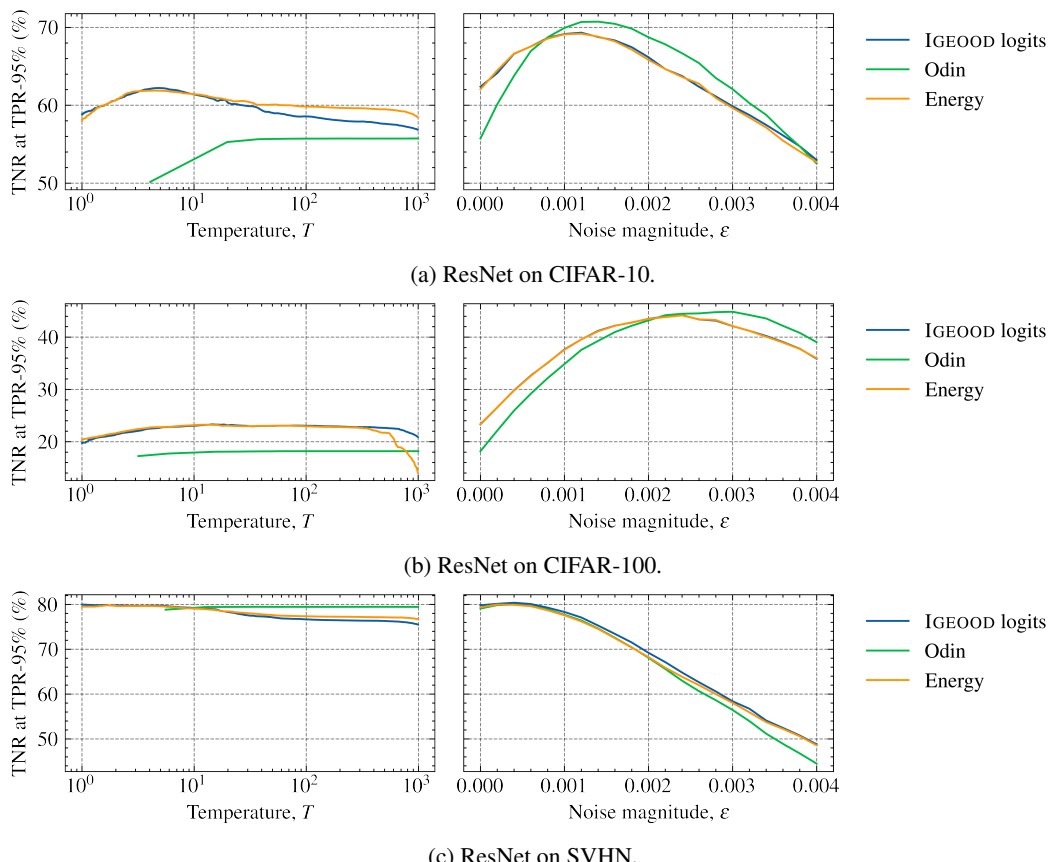

(a) ResNet on CIFAR-10.

(b) ResNet on CIFAR-100.

(c) ResNet on SVHN.

Figure 6: Temperature and noise magnitude tuning for OOD detection performance for ODIN (Liang et al., 2018), Energy (Liu et al., 2020) and IGEOOD (ours) on iSUN (Xu et al., 2015) OOD dataset for a ResNet-34 architecture.

Table 8: BLACK-BOX and GREY-BOX settings average performance across different OOD datasets for validation. The hyperparameters are tuned using one validation dataset (column 1), and evaluation is done on the remaining eight OOD test datasets. The DNN is DenseNet-BC-100 pre-trained on CIFAR-10, and the values are TNR at TPR-95% in percentage.

| | | BLACK-BOX | | | GREY-BOX | | |
|---|---|---|---|---|---|---|---|
| Validation set | Baseline | ODIN | Energy | IGEOOD | ODIN | Energy | IGEOOD |
| iSUN | 52.5 | 64.3 | 64.9 | **65.6** | **66.8** | 64.8 | 65.3 |
| Chars | 55.0 | 70.8 | 71.1 | **71.4** | 72.5 | 72.0 | **73.4** |
| CIFAR-100 | 55.4 | 68.6 | 69.1 | **72.0** | 68.6 | **71.7** | 71.3 |
| Gaussian | 49.4 | 62.8 | **65.6** | 63.4 | **70.4** | 64.0 | 68.0 |
| TinyImgNet | 53.0 | 64.7 | **65.2** | 63.5 | **67.0** | 65.0 | 65.5 |
| LSUN | 52.1 | **63.9** | 63.7 | 63.6 | **66.6** | 65.3 | 65.0 |
| Places365 | 55.3 | 68.5 | 69.0 | **71.8** | 70.0 | **71.5** | 70.9 |
| SVHN | 55.4 | 68.7 | 69.3 | **69.5** | 70.0 | 69.4 | **70.1** |
| Textures | 55.4 | 71.2 | **73.1** | 71.4 | 71.5 | **72.4** | 71.6 |
| average and std. | 53.7±2.0 | 67.1±3.0 | 67.9±3.0 | **68.0**±3.7 | **69.3**±2.0 | 68.4±3.4 | 69.0±3.0 |

Table 9: Experiment using five different training seeds for DenseNet-100 on CIFAR-10 for the BLACK-BOX scenario. The average test accuracy of the 5 models is 94.58%±0.13%. All values are percentages.

| Dataset | TNR at TPR-95% | AUROC |
|---|---|---|
| | Baseline / ODIN / Energy / IGEOOD | |
| Chars | $34.1_{\pm21}/54.3_{\pm20}/59.0_{\pm14}/\mathbf{59.4}_{\pm14}$ | $88.6_{\pm4.3}/90.4_{\pm3.5}/90.2_{\pm3.9}/\mathbf{90.6}_{\pm3.4}$ |
| CIFAR-100 | $37.1_{\pm0.5}/\mathbf{47.8}_{\pm1.4}/44.8_{\pm2.3}/45.2_{\pm2.2}$ | $88.2_{\pm0.3}/\mathbf{88.8}_{\pm0.7}/88.0_{\pm1.0}/88.3_{\pm0.8}$ |
| Gaussian | $42.7_{\pm49}/74.0_{\pm28}/79.6_{\pm23}/\mathbf{80.2}_{\pm23}$ | $93.7_{\pm4.3}/96.6_{\pm2.4}/\mathbf{96.7}_{\pm1.8}/\mathbf{96.7}_{\pm1.8}$ |
| TinyImgNet | $50.6_{\pm4.3}/76.1_{\pm4.4}/78.3_{\pm5.0}/\mathbf{78.4}_{\pm5.0}$ | $92.5_{\pm1.0}/95.7_{\pm1.0}/96.0_{\pm1.1}/\mathbf{96.1}_{\pm1.0}$ |
| LSUN | $58.0_{\pm3.6}/85.2_{\pm3.5}/87.3_{\pm4.7}/\mathbf{87.5}_{\pm4.5}$ | $94.2_{\pm0.6}/97.4_{\pm0.6}/97.6_{\pm0.7}/\mathbf{97.7}_{\pm0.7}$ |
| Places365 | $9.30_{\pm1.5}/\mathbf{54.2}_{\pm2.5}/52.7_{\pm4.2}/53.2_{\pm3.9}$ | $88.4_{\pm0.4}/\mathbf{89.9}_{\pm1.2}/89.4_{\pm1.7}/89.7_{\pm1.5}$ |
| SVHN | $36.0_{\pm3.0}/\mathbf{48.1}_{\pm7.1}/46.1_{\pm9.9}/46.5_{\pm9.6}$ | $\mathbf{86.8}_{\pm2.0}/86.6_{\pm4.8}/85.9_{\pm5.7}/86.4_{\pm5.0}$ |
| Textures | $35.6_{\pm1.7}/\mathbf{38.2}_{\pm1.4}/33.3_{\pm2.8}/34.0_{\pm2.6}$ | $\mathbf{87.2}_{\pm0.6}/83.3_{\pm1.0}/80.8_{\pm1.9}/82.1_{\pm1.3}$ |
| average and std. | $41.7_{\pm10}/59.7_{\pm8.6}/60.1_{\pm8.3}/\mathbf{60.6}_{\pm8.0}$ | $90.0_{\pm1.7}/\mathbf{91.1}_{\pm1.9}/90.6_{\pm2.2}/90.9_{\pm1.9}$ |

Table 10: Average and standard deviation OOD detection performance across eight OOD datasets for each model and in-distribution dataset in a GREY-BOX setting.

| Model | In-dist. | TNR at TPR-95% | AUROC |
|---|---|---|---|
| | | ODIN / Energy / IGEOOD | |
| DenseNet | C-10 | $\mathbf{66.8}_{\pm23}/64.8_{\pm25}/65.3_{\pm24}$ | $\mathbf{91.9}_{\pm6.2}/91.5_{\pm6.4}/\mathbf{91.9}_{\pm6.0}$ |
| | C-100 | $\mathbf{25.5}_{\pm14}/24.8_{\pm13}/25.0_{\pm13}$ | $76.6_{\pm12}/76.4_{\pm12}/\mathbf{78.2}_{\pm8.2}$ |
| | SVHN | $\mathbf{75.4}_{\pm15}/70.6_{\pm17}/72.4_{\pm16}$ | $\mathbf{91.6}_{\pm5.4}/89.2_{\pm6.9}/90.0_{\pm6.3}$ |
| ResNet | C-10 | $57.3_{\pm20}/57.7_{\pm19}/\mathbf{57.8}_{\pm19}$ | $\mathbf{89.2}_{\pm5.4}/88.7_{\pm5.3}/89.0_{\pm5.2}$ |
| | C-100 | $\mathbf{31.1}_{\pm22}/30.2_{\pm22}/30.2_{\pm22}$ | $\mathbf{76.9}_{\pm11}/74.4_{\pm12}/74.3_{\pm12}$ |
| | SVHN | $78.5_{\pm7.8}/78.5_{\pm7.9}/\mathbf{78.8}_{\pm7.8}$ | $90.4_{\pm3.4}/\mathbf{90.9}_{\pm3.4}/90.7_{\pm3.3}$ |
| Average and Std. | | $\mathbf{55.8}_{\pm21}/54.4_{\pm20}/54.9_{\pm20}$ | $\mathbf{86.1}_{\pm6.7}/85.2_{\pm7.0}/85.7_{\pm6.8}$ |

### E.6 IGEOOD COMPARED TO OTHER WHITE-BOX METHODS.

Even though Lee et al. (2018) shares the closest setup to ours, recent literature also shows promising results for OOD detection in a WHITE-BOX setting, achieving state-of-the-art in a few benchmarks. Notably, the works from Sastry & Oore (2020); Hsu et al. (2020); Zisselman & Tamar (2020) achieve remarkable performance in a range of benchmarks. Thus, we gathered the reported results from the original works and displayed them in Table 11 and 12, which considers that a few OOD samples and only adversarial samples are available for tuning, respectively. We highlight that Sastry & Oore (2020) extracts, in addition to the outputs of the blocks, intra-block features for the ResNet and DenseNet models.

### E.7 EXTENDED OOD DETECTION RESULTS

We show in Table 13 extended OOD detection results of Table 1. It contains the OOD detection performance for each model, in-distribution dataset and OOD dataset in a BLACK-BOX setting. In Table 14, we show the performance of ODIN, energy-based, and IGEOOD scores in the task of OOD detection in a GREY-BOX setup for each OOD dataset. In Table 15 and 16, we show additional results referring to the right-hand column and left-hand column of Table 2, respectively.

## F HISTOGRAMS

Figures 7, 8, 10 and 9 display histograms for the OOD detection score for IGEOOD in the BLOCK-BOX, GREY-BOX and WHITE-BOX settings, respectively.

Table 11: TNR at TPR-95% (%) performance in a WHITE-BOX setting considering the original results from Lee et al. (2018) and Zisselman & Tamar (2020) with access to OOD samples. The models are DenseNet-BC-100 and ResNet-34 pre-trained on CIFAR-10, CIFAR-100 and SVHN.

| | OOD dataset | CIFAR-10 | CIFAR-100 | SVHN |
|---|---|---|---|---|
| | | Mahalanobis / Res-Flow / IGEOOD / IGEOOD+ | | |
| DenseNet | iSUN | 95.3/ - /97.7/**99.8** | 87.0/ - /93.8/**99.7** | **99.9**/ - /98.3/**99.9** |
| | LSUN | 97.2/98.2/98.5/**99.9** | 91.4/96.3/95.2/**99.9** | **99.9**/100/97.1/**99.9** |
| | TinyImgNet | 95.0/96.4/95.7/**99.8** | 86.6/93.0/94.5/**99.5** | **99.9**/100/98.2/**99.9** |
| | SVHN/C-10 | 90.8/94.9/98.9/**99.9** | 82.5/84.9/93.3/**99.6** | 96.8/**99.0**/91.6/98.3 |
| | average | 94.6/96.5/97.7/**99.8** | 86.9/91.4/94.2/**99.7** | 99.1/**99.6**/96.3/**99.5** |
| ResNet | iSUN | 97.8/ - /97.2/**99.9** | 89.9/ - /93.4/**99.8** | 99.7/ - /99.8/**100** |
| | LSUN | 98.8/99.0/98.4/**100** | 90.9/96.2/94.3/**100** | **99.9**/100/99.7/**99.9** |
| | TinyImgNet | 97.1/97.8/96.3/**99.6** | 90.9/94.6/90.1/**99.6** | **99.9**/100/99.7/**99.9** |
| | SVHN/C-10 | 87.8/96.5/98.8/**99.8** | 91.9/93.0/91.6/**99.7** | 98.4/99.4/97.7/**99.7** |
| | average | 95.4/97.8/97.7/**99.8** | 90.9/94.6/92.35/**99.8** | 99.5/99.8/99.2/**99.9** |

Table 12: TNR at TPR-95% (%) performance in a WHITE-BOX setting considering the original results from Lee et al. (2018); Sastry & Oore (2020); Hsu et al. (2020); Zisselman & Tamar (2020) without access to OOD samples for hyperparameter tuning.

| | OOD dataset | CIFAR-10 | CIFAR-100 | SVHN |
|---|---|---|---|---|
| | | Mahalanobis / Gram Matrix / DeConf-C / Res-Flow / IGEOOD / IGEOOD+ | | |
| DenseNet | iSUN | 94.3/99.0/**99.4**/ - /94.5/95.8 | 84.8/95.9/**98.4**/ - /93.8/92.2 | **99.9**/99.4/ - / - /98.2/98.6 |
| | LSUN | 97.2/**99.5**/99.4/98.1/96.4/97.2 | 91.4/97.2/**98.7**/95.8/95.1/94.4 | **100**/99.5/ - /**100**/97.3/97.0 |
| | TinyImgNet | 94.9/98.8/**99.1**/96.1/93.4/94.5 | 87.2/95.7/**98.6**/91.5/94.3/94.0 | **99.9**/99.1/ - /**99.9**/98.1/96.8 |
| | SVHN/C-10 | 89.9/96.1/**98.8**/86.1/94.3/95.7 | 62.2/89.3/**95.9**/48.9/90.1/90.6 | **90.0**/80.4/ - /**90.0**/89.5/86.6 |
| | average | 94.1/98.3/**99.2**/93.4/94.6/95.8 | 81.4/94.5/**97.9**/78.7/93.3/92.8 | **97.4**/94.6/ - /96.6/95.8/94.8 |
| ResNet | iSUN | 96.8/**99.3**/88.8/ - /95.3/95.0 | 87.9/**94.8**/75.3/ - /89.4/91.0 | **100**/99.4/ - / - /99.8/99.9 |
| | LSUN | 98.1/**99.6**/90.9/99.1/97.7/97.7 | 56.6/**96.6**/76.8/70.4/88.6/93.9 | 99.9/99.6/ - /**100**/99.8/100 |
| | TinyImgNet | 95.5/**98.7**/81.4/98.0/94.3/94.2 | 70.3/**94.8**/76.5/77.5/86.2/90.1 | 99.2/99.3/ - /**99.9**/99.6/99.6 |
| | SVHN/C-10 | 75.8/97.6/89.5/91.0/**98.2**/97.7 | 41.9/**80.8**/55.1/74.1/75.2/78.5 | 94.1/85.8/ - /96.6/96.7/**97.3** |
| | average | 91.5/**98.8**/87.6/96.0/96.3/96.2 | 64.2/**91.7**/71.0/74.0/84.8/88.4 | 98.3/96.0/ - /98.8/99.0/**99.2** |

Table 13: Extended BLACK-BOX results for Table1. Parameter tuning on iSUN dataset.

| In-dist. (model) | OOD dataset | TNR at TPR-95% | AUROC | AUPR |
|---|---|---|---|---|
| | | Baseline / ODIN / Energy / IGEOOD | | |
| CIFAR-10 (DenseNet) | Chars | 43.5/**57.2**/54.6/55.0 | 90.2/**91.2**/90.4/90.5 | 93.0/**93.1**/92.5/92.7 |
| | CIFAR-100 | 40.6/**53.1**/50.5/50.7 | 89.4/**90.4**/89.7/89.8 | 90.5/**90.7**/90.1/90.2 |
| | Gaussian | 88.1/99.8/**99.9**/**99.9** | 97.6/**98.9**/98.5/98.5 | 98.3/**99.3**/99.1/99.1 |
| | TinyImgNet | 59.4/85.0/**88.0**/87.8 | 94.1/97.3/**97.6**/97.6 | 95.4/97.7/**97.9**/97.9 |
| | LSUN | 66.9/91.4/**93.3**/93.3 | 95.5/98.3/**98.5**/98.5 | 96.5/98.5/**98.7**/98.7 |
| | Places365 | 40.8/**54.2**/51.5/52.0 | 88.8/**90.2**/89.5/89.7 | 74.4/**74.8**/73.9/74.3 |
| | SVHN | 40.4/**52.0**/49.6/50.1 | 89.9/**90.9**/90.2/90.3 | **84.6**/84.6/83.5/83.7 |
| | Textures | 40.5/**42.1**/34.9/35.6 | **88.5**/85.1/82.4/83.2 | **93.1**/88.3/86.5/87.8 |
| | average | 52.5/**66.8**/65.3/65.6 | 91.7/**92.8**/92.1/92.3 | 90.7/**90.9**/90.3/90.6 |
| CIFAR-100 (DenseNet) | Chars | 15.1/**17.8**/17.0/17.2 | 72.8/**78.0**/77.9/77.8 | 79.6/83.8/**83.9**/83.8 |
| | CIFAR-10 | 17.7/**18.1**/17.1/17.0 | **75.6**/74.8/74.4/75.4 | **78.3**/74.3/74.0/76.2 |
| | Gaussian | 0.0/0.0/0.0/0.0 | 30.2/19.4/19.5/**30.7** | 53.2/44.0/44.1/**53.5** |
| | TinyImgNet | 16.7/24.7/25.0/**25.7** | 72.0/79.4/**79.6**/79.5 | 74.8/80.7/**80.9**/80.7 |
| | LSUN | 15.5/23.2/24.2/**25.4** | 70.9/80.4/**80.8**/80.6 | 74.4/82.6/**82.9**/82.5 |
| | Places365 | 18.8/21.2/**20.6**/20.6 | 75.9/**78.0**/77.7/78.0 | 54.2/54.5/54.3/**55.7** |
| | SVHN | 25.7/36.4/36.5/**36.7** | 82.8/**88.4**/88.4/88.2 | 75.4/**82.5**/82.4/82.4 |
| | Textures | 18.0/22.4/22.2/**22.8** | 72.7/74.4/74.3/**75.7** | 80.8/79.1/79.0/**81.5** |
| | average | 15.9/20.5/20.3/**20.7** | 69.1/71.6/71.6/**73.2** | 71.3/72.7/72.7/**74.5** |
| SVHN (DenseNet) | Chars | 46.4/27.0/45.0/**47.7** | **83.9**/52.2/79.6/80.9 | **91.8**/70.2/88.9/89.6 |
| | CIFAR-10 | 61.8/66.0/64.7/**67.0** | **92.3**/90.9/90.3/90.9 | **96.2**/95.1/94.6/95.0 |
| | CIFAR-100 | 61.3/64.4/63.0/**65.4** | **91.9**/90.3/89.5/90.3 | **95.7**/94.3/93.8/94.3 |
| | Gaussian | 93.6/97.8/97.9/**98.0** | 97.4/**98.0**/98.0/98.0 | 99.2/**99.4**/99.4/99.4 |
| | TinyImgNet | 80.4/84.4/84.1/**85.1** | **95.5**/95.3/94.9/95.3 | **97.9**/97.4/97.2/97.5 |
| | LSUN | 80.1/84.4/84.3/**85.0** | **95.5**/95.3/95.1/95.3 | **98.0**/97.6/97.5/97.6 |
| | Places365 | 66.8/71.0/69.9/**71.9** | **93.0**/91.9/91.3/91.9 | **89.1**/85.8/84.6/85.8 |
| | Textures | 56.4/55.2/52.4/**56.5** | **88.9**/84.6/82.5/84.9 | **95.5**/93.3/92.2/93.4 |
| | average | 68.3/68.8/70.2/**72.1** | **92.3**/87.3/90.2/90.9 | **95.4**/91.6/93.5/94.1 |
| CIFAR-10 (ResNet) | Chars | 36.8/45.8/50.7/**51.1** | 89.4/90.1/90.3/**90.4** | **92.7**/92.7/92.5/92.7 |
| | CIFAR-100 | 33.6/41.8/45.6/**45.9** | 86.4/87.0/87.0/**87.1** | **87.0**/86.6/86.2/86.4 |
| | Gaussian | 81.5/**89.9**/88.2/89.0 | 96.9/**97.3**/96.7/96.7 | 97.9/**98.3**/98.0/98.0 |
| | TinyImgNet | 42.1/53.4/57.7/**58.2** | 90.3/91.5/91.6/**91.7** | 91.8/**92.2**/92.1/92.2 |
| | LSUN | 41.2/55.1/61.7/**62.0** | 90.1/91.5/92.0/**92.1** | 91.5/92.1/**92.3**/92.3 |
| | Places365 | 32.9/42.4/**48.2**/48.2 | 85.8/86.6/**86.9**/86.9 | **67.1**/66.1/65.6/65.7 |
| | SVHN | 27.7/39.9/48.6/**48.8** | 89.2/90.2/90.5/**90.6** | **85.8**/85.1/84.1/84.5 |
| | Textures | 37.9/46.6/49.6/**50.3** | 89.0/**89.1**/88.4/88.6 | **93.9**/93.3/92.4/92.6 |
| | average | 41.7/51.9/56.3/**56.7** | 89.6/90.4/90.4/**90.5** | **88.5**/88.3/87.9/88.1 |
| CIFAR-100 (ResNet) | Chars | 14.3/**15.3**/15.1/15.2 | 72.7/73.0/73.1/**73.4** | **77.8**/77.2/77.3/77.7 |
| | CIFAR-10 | 18.1/**18.4**/17.4/18.0 | 76.5/76.6/76.5/**76.8** | **78.3**/77.8/77.8/78.1 |
| | Gaussian | **1.6**/0.8/0.3/0.6 | 72.8/76.0/**76.7**/76.7 | 81.6/83.8/**84.4**/84.3 |
| | TinyImgNet | 17.8/20.7/**23.8**/23.0 | 73.3/76.6/**77.4**/76.9 | 76.4/78.7/**79.2**/78.8 |
| | LSUN | 15.5/18.8/**21.5**/20.6 | 70.8/74.1/**74.9**/74.4 | 72.9/75.2/**75.7**/75.2 |
| | Places365 | **17.3**/17.2/16.2/16.9 | **74.1**/73.2/72.9/73.4 | **44.6**/42.1/41.9/42.7 |
| | SVHN | **14.2**/13.8/12.5/13.3 | **74.8**/74.1/73.9/74.5 | **59.4**/56.9/56.8/57.7 |
| | Textures | 20.9/22.8/**23.3**/23.3 | 77.0/78.0/78.2/**78.3** | 85.9/86.2/86.3/**86.4** |
| | average | 15.0/16.0/16.3/**16.4** | 74.0/75.2/75.4/**75.5** | 72.1/72.3/72.4/**72.6** |
| SVHN (ResNet) | Chars | 56.9/58.2/58.4/**58.5** | **85.1**/83.7/83.7/84.0 | **92.3**/91.0/91.0/91.3 |
| | CIFAR-10 | 79.0/80.6/81.0/**81.3** | **93.0**/92.1/92.2/92.5 | **94.7**/93.4/93.4/93.7 |
| | CIFAR-100 | 78.0/79.6/79.8/**80.2** | **92.7**/91.9/91.9/92.2 | **94.7**/93.4/93.4/93.7 |
| | Gaussian | 85.3/86.7/86.9/**87.3** | **95.9**/95.7/95.7/95.9 | **97.7**/97.1/97.1/97.3 |
| | TinyImgNet | 79.8/81.4/81.8/**82.2** | **93.5**/92.9/92.9/93.2 | **95.3**/94.3/94.3/94.5 |
| | LSUN | **74.9**/76.8/77.2/77.4 | **91.5**/90.5/90.6/90.9 | **93.5**/92.2/92.1/92.4 |
| | Places365 | 77.0/78.4/78.8/**79.0** | **92.0**/91.0/91.1/91.4 | **81.4**/77.9/77.8/78.6 |
| | Textures | 78.5/80.0/80.4/**80.9** | **93.7**/93.0/93.0/93.4 | **97.6**/97.0/97.0/97.2 |
| | average | 76.2/77.7/78.0/**78.4** | **92.2**/91.3/91.4/91.7 | **93.4**/92.0/92.0/92.4 |
| Average of the average values | | 46.0/51.6/52.1/**52.6** | 85.3/85.4/85.1/**85.8** | **85.2**/84.7/84.4/85.1 |

Table 14: Extended GREY-BOX results. Parameter tuning on iSUN dataset.

| In-dist. (model) | OOD dataset | TNR at TPR-95% | AUROC | AUPR |
|---|---|---|---|---|
| | | ODIN / Energy / IGEOOD | | |
| CIFAR-10 (DenseNet) | Chars | **52.5**/49.2/50.0 | **89.1**/88.2/88.7 | 90.7/90.2/**90.8** |
| | CIFAR-100 | 48.9/49.7/**50.0** | 88.3/88.5/**88.7** | 88.4/88.6/**88.9** |
| | Gaussian | **100/100/100** | **100**/99.9/99.9 | **100**/99.9/99.9 |
| | TinyImgNet | **92.6**/92.4/92.4 | 98.5/98.5/98.5 | **98.6**/98.5/98.5 |
| | LSUN | 96.2/96.2/96.2 | **99.2**/99.1/**99.2** | **99.3**/99.2/99.2 |
| | Places365 | **52.4**/52.0/52.4 | 89.0/88.9/**89.2** | 88.7/88.7/**89.0** |
| | SVHN | **49.3**/41.8/42.5 | **89.7**/88.2/88.6 | **82.1**/80.6/81.4 |
| | Textures | **42.9**/37.0/38.7 | 81.4/80.8/**82.3** | 84.7/84.6/**86.5** |
| | average | **66.8**/64.8/65.3 | **91.9**/91.5/**91.9** | 91.6/91.3/**91.8** |
| CIFAR-100 (DenseNet) | Chars | **19.8**/19.2/19.5 | **76.6**/76.3/75.8 | **80.8**/80.6/79.7 |
| | CIFAR-10 | 15.3/16.4/**16.9** | 72.4/72.8/**74.9** | 72.6/72.8/**76.1** |
| | Gaussian | 0.0/0.0/0.0 | 47.1/46.9/**59.9** | 65.5/65.4/**74.9** |
| | TinyImgNet | **43.8**/42.2/40.1 | **86.5**/86.2/84.8 | **87.2**/87.0/85.3 |
| | LSUN | **42.2**/40.6/38.9 | **86.8**/86.4/84.7 | **87.8**/87.7/85.7 |
| | Places365 | 23.4/23.8/**23.9** | 79.1/79.1/**79.4** | 79.4/79.4/**80.0** |
| | SVHN | 34.7/31.4/**35.7** | 87.7/87.2/**88.1** | 81.0/80.8/**81.8** |
| | Textures | **24.9**/24.7/**24.9** | 76.7/76.6/**77.8** | 81.7/81.6/**83.8** |
| | average | **25.5**/24.8/25.0 | 76.6/76.4/**78.2** | 79.5/79.4/**80.9** |
| SVHN (DenseNet) | Chars | **53.3**/45.8/48.3 | **81.8**/78.0/79.3 | **90.4**/88.1/88.8 |
| | CIFAR-10 | **69.8**/65.1/67.1 | **91.3**/89.1/89.9 | **95.2**/93.7/94.2 |
| | CIFAR-100 | **69.8**/64.2/66.3 | **91.1**/88.5/89.4 | **94.8**/93.0/93.6 |
| | Gaussian | **99.4**/99.0/99.1 | **98.9**/98.5/98.6 | **99.7**/99.5/99.6 |
| | TinyImgNet | **88.4**/85.5/86.4 | **96.1**/94.9/95.2 | **97.9**/97.0/97.3 |
| | LSUN | **88.2**/85.8/86.5 | **96.2**/95.1/95.4 | **98.0**/97.3/97.5 |
| | Places365 | **74.8**/69.9/71.8 | **92.4**/90.3/91.0 | **95.8**/94.2/94.7 |
| | Textures | **59.4**/49.6/53.7 | **84.9**/79.0/81.4 | **93.4**/90.4/91.6 |
| | average | **75.4**/70.6/72.4 | **91.6**/89.2/90.0 | **95.7**/94.2/94.7 |
| CIFAR-10 (ResNet) | Chars | 54.7/54.2/**54.8** | **88.3**/87.8/88.2 | **90.7**/90.4/90.6 |
| | CIFAR-100 | 38.9/**41.7**/41.5 | 83.8/83.8/**84.1** | 83.7/83.5/**83.8** |
| | Gaussian | **100/100/100** | **100**/99.5/99.5 | **100**/99.7/99.7 |
| | TinyImgNet | **68.7**/66.6/67.0 | **93.1**/92.4/92.6 | **93.3**/92.7/92.9 |
| | LSUN | **70.4**/68.8/68.9 | **93.2**/92.8/92.9 | **93.3**/92.9/93.0 |
| | Places365 | 40.4/**43.2**/43.0 | 84.1/84.1/**84.2** | 84.0/83.9/**84.0** |
| | SVHN | 38.6/39.9/**40.0** | **84.8**/84.0/84.6 | **76.4**/74.6/75.7 |
| | Textures | 46.8/47.1/**47.5** | **86.1**/85.2/85.7 | **91.4**/90.7/91.0 |
| | average | 57.3/57.7/**57.8** | **89.2**/88.7/89.0 | **89.1**/88.5/88.8 |
| CIFAR-100 (ResNet) | Chars | 14.8/13.8/13.8 | 68.0/65.1/65.0 | 71.0/68.5/68.4 |
| | CIFAR-10 | **14.3**/11.8/11.8 | **70.8**/65.1/64.9 | **70.1**/64.7/64.5 |
| | Gaussian | **78.7**/73.8/74.3 | **96.8**/95.9/96.0 | **97.7**/97.4/97.4 |
| | TinyImgNet | 44.9/49.2/**49.3** | 86.4/**86.5/86.5** | **86.1**/85.6/85.6 |
| | LSUN | 40.2/44.3/**44.4** | 83.5/**83.8/83.8** | 82.5/82.5/82.5 |
| | Places365 | **15.9**/10.2/10.1 | **67.7**/59.7/59.5 | **64.5**/58.2/58.0 |
| | SVHN | 10.2/**10.6/10.6** | 64.3/**65.4**/65.3 | 42.2/**43.6**/43.5 |
| | Textures | **29.5**/27.6/27.7 | **77.9**/73.7/73.6 | **84.4**/81.1/81.0 |
| | average | **31.1**/30.2/30.2 | **76.9**/74.4/74.3 | **74.8**/72.7/72.6 |
| SVHN (ResNet) | Chars | 59.9/59.4/**60.0** | 82.8/**83.2/83.2** | 90.4/**90.7**/90.6 |
| | CIFAR-10 | 80.9/81.0/**81.3** | 90.9/**91.5**/91.2 | 92.2/**92.8**/92.5 |
| | CIFAR-100 | 80.2/80.2/**80.5** | 90.8/**91.4**/91.1 | 92.3/**92.9**/92.6 |
| | Gaussian | **89.4**/88.7/**89.4** | 95.9/95.9/95.9 | **97.3**/97.2/97.2 |
| | TinyImgNet | 82.6/82.3/**82.9** | 92.2/**92.6**/92.5 | 93.5/**93.9**/93.8 |
| | LSUN | 77.8/77.6/**78.3** | 89.6/**90.1**/89.9 | 91.3/**91.7**/91.6 |
| | Places365 | 78.7/78.8/**79.0** | 89.8/**90.6**/90.2 | 91.2/**91.9**/91.5 |
| | Textures | 78.6/**79.8**/79.2 | 91.3/**92.2**/91.7 | 96.1/**96.6**/96.3 |
| | average | 78.5/78.5/**78.8** | 90.4/**90.9**/90.7 | 93.0/**93.5**/93.3 |
| Average of average values | | **55.8**/54.4/54.9 | **86.1**/85.2/85.7 | **87.3**/86.6/87.0 |

Table 15: WHITE-BOX extended results. Validation on OOD data.

| In-dist. (model) | OOD dataset | TNR at TPR-95% | AUROC | AUPR |
|---|---|---|---|---|
| | | Mahalanobis (Lee et al., 2018) / IGEOOD+ | | |
| CIFAR-10 (DenseNet) | Chars | 91.3/**99.4** | 97.5/**99.9** | 97.7/**99.9** |
| | CIFAR-100 | 21.4/**56.6** | 67.3/**90.7** | 64.4/**90.8** |
| | TinyImgNet | 96.9/**99.8** | 99.3/**99.9** | 99.3/**99.9** |
| | LSUN | 98.2/**99.9** | 99.5/**100** | 99.5/**100** |
| | Places365 | 18.1/**80.2** | 72.7/**95.7** | 72.8/**95.4** |
| | SVHN | 90.1/**99.9** | 97.3/**100** | 97.3/**100** |
| | Textures | 84.1/**97.4** | 95.6/**99.5** | 94.7/**99.5** |
| | Gaussian | **100/100** | **100/100** | **100/100** |
| | iSUN | 97.3/**99.8** | 99.4/**100** | 99.4/**100** |
| | average | 77.5±31/**92.6**±14 | 92.1±12/**98.4**±3.0 | 91.7±13/**98.4**±3.0 |
| CIFAR-100 (DenseNet) | Chars | 62.9/**97.5** | 94.0/**99.4** | 95.8/**99.4** |
| | CIFAR-10 | 9.1/**22.7** | 60.8/**80.7** | 60.1/**83.0** |
| | TinyImgNet | 87.1/**99.5** | 97.4/**99.9** | 97.4/**99.9** |
| | LSUN | 91.1/**99.9** | 97.8/**100** | 98.1/**100** |
| | Places365 | 5.9/**58.2** | 54.8/**90.0** | 54.7/**89.2** |
| | SVHN | 79.0/**99.6** | 96.8/**99.9** | 94.1/**99.9** |
| | Textures | 70.3/**90.2** | 91.4/**98.1** | 94.3/**98.2** |
| | Gaussian | **100/100** | **100/100** | **100/100** |
| | iSUN | 86.4/**99.7** | 96.8/**99.9** | 97.7/**99.9** |
| | average | 67.7±28/**90.2**±21 | 87.8±13/**97.7**±5.0 | 88.0±12/**97.8**±5.0 |
| SVHN (DenseNet) | Chars | 78.7/**92.2** | 96.1/**98.4** | **98.9**/98.5 |
| | CIFAR-10 | 91.6/**98.3** | 98.0/**99.6** | 99.4/**99.6** |
| | CIFAR-100 | 92.9/**95.3** | 98.2/**99.1** | **99.4**/99.2 |
| | TinyImgNet | **99.9/99.9** | 99.8/**99.9** | **99.9/99.9** |
| | LSUN | **99.9/99.9** | 99.8/**100** | 99.7/**100** |
| | Places365 | 94.7/**98.3** | 98.3/**99.6** | 98.4/**99.7** |
| | Textures | 98.2/**98.5** | 99.4/**99.6** | **99.9**/99.6 |
| | Gaussian | **100/100** | **100/100** | **100/100** |
| | iSUN | **99.9/99.9** | 99.8/**99.9** | **99.9/99.9** |
| | average | 95.1±8.0/**98.0**±2.0 | 98.8±1.0/**99.6**±0.1 | 99.5±1.0/**99.6**±0.1 |
| CIFAR-10 (ResNet) | Chars | 93.6/**99.3** | 98.6/**99.8** | 99.1/**99.8** |
| | CIFAR-100 | 44.9/**51.3** | 87.4/**90.9** | 87.8/**91.7** |
| | TinyImgNet | 96.8/**99.6** | 99.4/**99.9** | 99.4/**99.9** |
| | LSUN | 98.3/**99.9** | 99.6/**100** | 99.6/**100** |
| | Places365 | 45.8/**77.6** | 88.1/**95.6** | 88.1/**95.5** |
| | SVHN | 96.1/**99.8** | 99.0/**99.9** | 98.1/**99.9** |
| | Textures | 84.3/**97.0** | 97.3/**99.4** | 98.6/**99.4** |
| | Gaussian | **100/100** | **100/100** | **100/100** |
| | iSUN | 97.2/**99.9** | 99.4/**100** | 99.5/**100** |
| | average | 84.1±23/**91.6**±16 | 96.5±4.0/**98.4**±3.0 | 96.7±4.0/**98.5**±3.0 |
| CIFAR-100 (ResNet) | Chars | 63.8/**97.8** | 94.0/**99.5** | 96.0/**99.5** |
| | CIFAR-10 | 18.0/**30.8** | 76.6/**85.3** | 76.4/**87.8** |
| | TinyImgNet | 90.1/**99.6** | 97.9/**99.9** | 98.0/**99.9** |
| | LSUN | 92.4/**100** | 98.3/**100** | 98.5/**100** |
| | Places365 | 23.5/**59.1** | 76.8/**91.2** | 76.0/**91.4** |
| | SVHN | 88.4/**99.7** | 97.7/**99.9** | 95.2/**99.9** |
| | Textures | 71.6/**90.7** | 93.9/**98.2** | 96.6/**98.1** |
| | Gaussian | **100/100** | **100/100** | **100/100** |
| | iSUN | 89.4/**99.8** | 97.7/**99.9** | 98.0/**99.9** |
| | average | 70.8±30/**86.4**±23 | 92.5±10/**97.1**±5.0 | 92.7±10/**97.4**±4.0 |
| SVHN (ResNet) | Chars | 84.9/**92.4** | 97.0/**98.4** | **99.0**/98.5 |
| | CIFAR-10 | 98.0/**99.7** | 99.2/**99.9** | 99.7/**99.9** |
| | CIFAR-100 | 98.3/**99.1** | 99.3/**99.7** | **99.8/99.8** |
| | TinyImgNet | **99.9/99.9** | 99.9/**100** | **100/100** |
| | LSUN | **99.9/99.9** | 99.9/**100** | **100/100** |
| | Places365 | 98.4/**99.6** | 99.3/**99.9** | 99.8/**99.9** |
| | Textures | 99.0/**99.9** | 99.7/**99.9** | **99.9/99.9** |
| | Gaussian | **100/100** | **100/100** | **100/100** |
| | iSUN | **100/100** | 99.9/**100** | **100/100** |
| | average | 97.6±6.0/**98.9**±2.0 | 99.4±1.0/**99.7**±0.1 | **99.8**±1.0/**99.8**±0.1 |
| Avg. and std. of avg. values | | 82.1±11/**92.9**±4.0 | 94.5±4.0/**98.5**±1.0 | 94.7±4.0/**98.6**±1.0 |

Table 16: WHITE-BOX extended results. Validation on adversarial (FGSM) data.

| In-dist. (model) | OOD dataset | TNR at TPR-95% | AUROC | AUPR |
|---|---|---|---|---|
| | | Mahalanobis (Lee et al., 2018) / IGEOOD | | |
| CIFAR-10 (DenseNet) | Chars | **88.5**/87.3 | **97.7**/97.7 | **98.3**/98.3 |
| | CIFAR-100 | 21.5/**26.4** | 68.0/**77.7** | 66.3/**75.5** |
| | TinyImageNet | **93.9**/93.4 | 98.6/**98.7** | 98.6/**98.7** |
| | LSUN | 96.3/**96.4** | 99.1/**99.2** | 99.1/**99.2** |
| | Places365 | 17.8/**23.2** | 70.0/**77.9** | 40.1/**76.3** |
| | SVHN | 87.0/**94.3** | 97.2/**98.7** | 93.7/**97.3** |
| | Textures | 83.6/**86.0** | 95.8/**97.2** | 97.3/**98.2** |
| | Gaussian | **100/100** | **100/100** | **100/100** |
| | iSUN | 94.3/**94.5** | 98.8/**98.9** | 98.9/**99.0** |
| | average | 75.9±30/**77.9**±29 | 91.7±12/**94.0**±9.0 | 88.0±20/**93.6**±10 |
| CIFAR-100 (DenseNet) | CIFAR-10 | 1.1/**5.7** | 43.5/**62.6** | 46.7/**62.6** |
| | Chars | 53.9/**59.6** | **92.2**/92.0 | **94.5**/93.9 |
| | TinyImageNet | 86.4/**94.3** | 97.4/**98.8** | 97.5/**98.9** |
| | LSUN | 88.6/**95.1** | 97.6/**98.9** | 97.9/**98.9** |
| | Places365 | 5.5/**13.0** | 56.6/**71.0** | 57.5/**71.0** |
| | SVHN | 56.1/**90.1** | 91.8/**98.0** | 85.4/**96.2** |
| | Textures | 67.5/**86.7** | 91.2/**97.4** | 94.4/**98.4** |
| | Gaussian | **100/100** | **100/100** | **100/100** |
| | iSUN | 84.8/**93.8** | 97.2/**98.7** | 97.6/**98.8** |
| | average | 60.4±34/**70.9**±35 | 85.3±19/**90.8**±13 | 85.7±19/**91.0**±13 |
| SVHN (DenseNet) | CIFAR-10 | **90.6**/89.5 | 97.7/**97.8** | 99.1/**99.2** |
| | CIFAR-100 | **91.8**/88.4 | **98.0**/97.7 | **99.2**/99.1 |
| | Chars | **72.3**/70.5 | **95.2**/94.5 | **98.5**/98.3 |
| | TinyImageNet | **99.5**/98.1 | **99.6**/99.3 | 99.5/**99.8** |
| | LSUN | **99.9**/97.3 | **99.8**/99.1 | **99.9**/99.7 |
| | Places365 | **94.3**/91.9 | **98.3**/98.2 | 98.1/**99.3** |
| | Textures | 95.3/**97.1** | 98.8/**99.3** | 99.6/**99.8** |
| | Gaussian | **100/100** | **100**/99.9 | **100/100** |
| | iSUN | **99.9**/98.2 | **99.8**/99.3 | **99.9**/99.8 |
| | average | **93.7**±8.0/92.3±9.0 | **98.6**±1.0/98.3±2.0 | 99.3±1.0/**99.4**±0.5 |
| CIFAR-10 (ResNet) | CIFAR-100 | **36.5**/21.5 | **84.5**/63.3 | **84.3**/58.1 |
| | Chars | 82.0/**90.9** | 96.9/**98.3** | 97.7/**98.7** |
| | TinyImageNet | **96.2**/94.3 | **99.2**/98.0 | **99.2**/96.7 |
| | LSUN | **98.2**/97.7 | **99.5**/99.2 | **99.5**/98.9 |
| | Places365 | **34.8**/15.9 | **85.0**/60.1 | **84.2**/24.4 |
| | SVHN | 81.0/**98.2** | 96.6/**99.3** | 93.7/**97.5** |
| | Textures | **81.7**/81.6 | **96.7**/93.4 | **98.2**/94.3 |
| | Gaussian | **100/100** | **100/100** | **100/100** |
| | iSUN | **96.8**/95.3 | **99.3**/98.6 | **99.3**/98.1 |
| | average | **78.6**±24/77.3±32 | **95.3**±6.0/90.0±15 | **95.1**±6.0/85.2±25 |
| CIFAR-100 (ResNet) | CIFAR-10 | 3.0/**5.0** | **61.0**/59.6 | **63.7**/60.6 |
| | Chars | 39.9/**55.1** | 85.6/**90.4** | 88.1/**92.5** |
| | TinyImageNet | **88.7**/86.2 | **97.6**/97.3 | **97.6**/97.3 |
| | LSUN | **91.3**/88.6 | **98.0**/97.8 | **98.3**/98.0 |
| | Places365 | 8.0/**8.6** | **67.9**/63.0 | **66.8**/61.7 |
| | SVHN | 31.6/**75.2** | 82.9/**95.8** | 68.8/**92.7** |
| | Textures | 65.9/**78.1** | 91.9/**95.6** | 95.2/**97.6** |
| | Gaussian | **100/100** | **100/100** | **100/100** |
| | iSUN | 87.9/**89.4** | 97.4/**97.8** | 97.6/**97.7** |
| | average | 57.4±36/**65.1**±33 | **86.9**±13/88.6±15 | 86.2±14/**88.7**±15 |
| SVHN (ResNet) | CIFAR-10 | **97.1**/96.7 | 99.1/**99.2** | 99.7/**99.7** |
| | CIFAR-100 | **97.5**/96.2 | **99.1**/99.1 | **99.7**/99.6 |
| | Chars | **75.4**/55.1 | **95.3**/89.1 | **98.5**/96.0 |
| | TinyImageNet | **99.9**/99.6 | **99.9/99.9** | **99.9/99.9** |
| | LSUN | **100**/99.8 | **99.9/99.9** | **100/100** |
| | Places365 | **98.1**/97.0 | **99.2/99.2** | **99.2**/99.0 |
| | Textures | **98.9**/98.4 | **99.6/99.6** | **99.9/99.9** |
| | Gaussian | **100/100** | 99.9/**100** | **100/100** |
| | iSUN | **100**/99.8 | 99.8/**99.9** | 99.9/**100** |
| | average | **96.3**±8.0/93.6±14 | **99.1**±1.0/98.4±3.0 | **99.6**±0.5/99.3±1.0 |
| Avg. and std. of avg. values | | 77.0±15/**79.5**±10 | 92.8±5.4/**93.4**±3.9 | 92.3±5.9/**92.9**±5.2 |

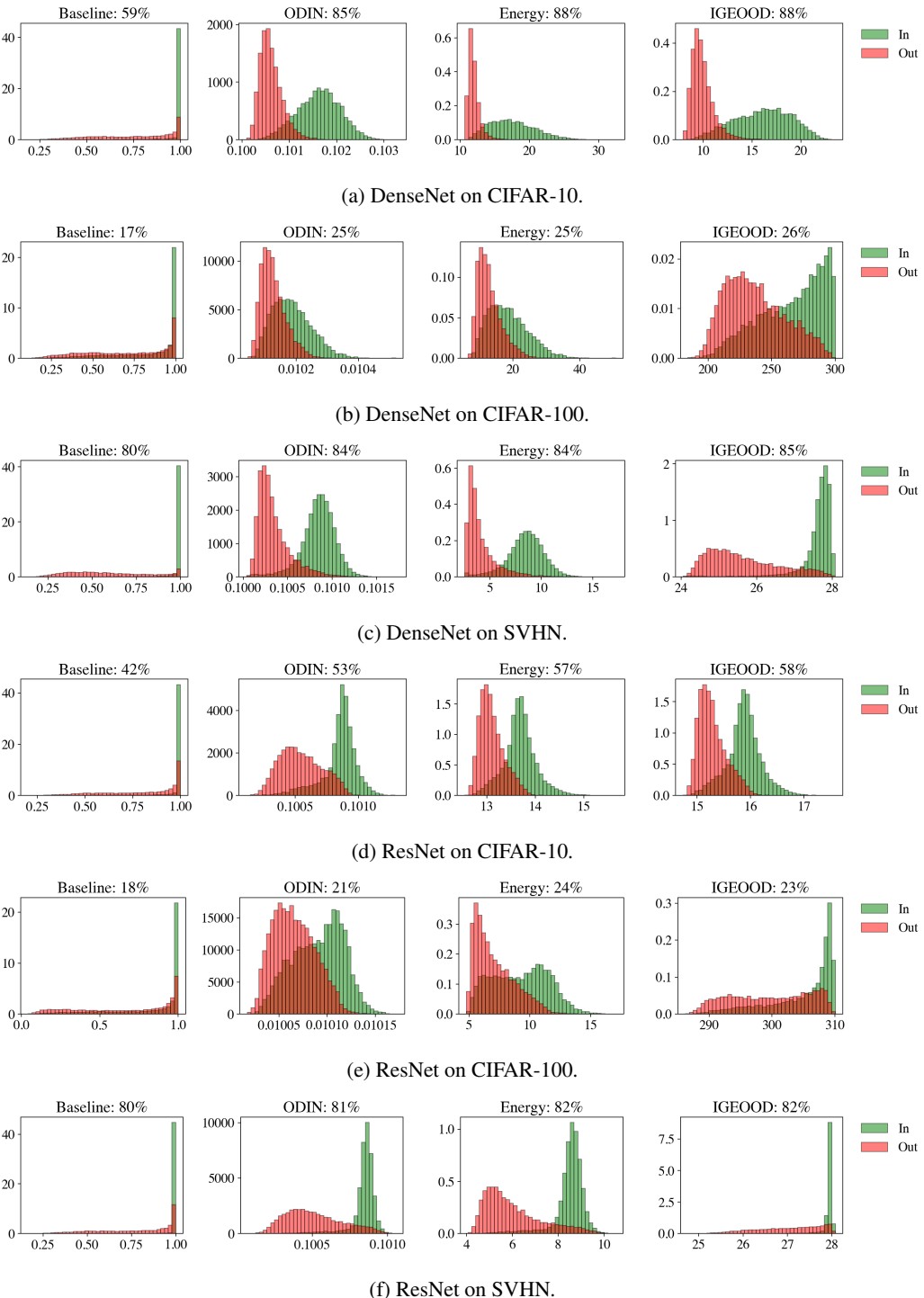

Figure 7: BLACK-BOX setup. TinyImageNet as OOD dataset.

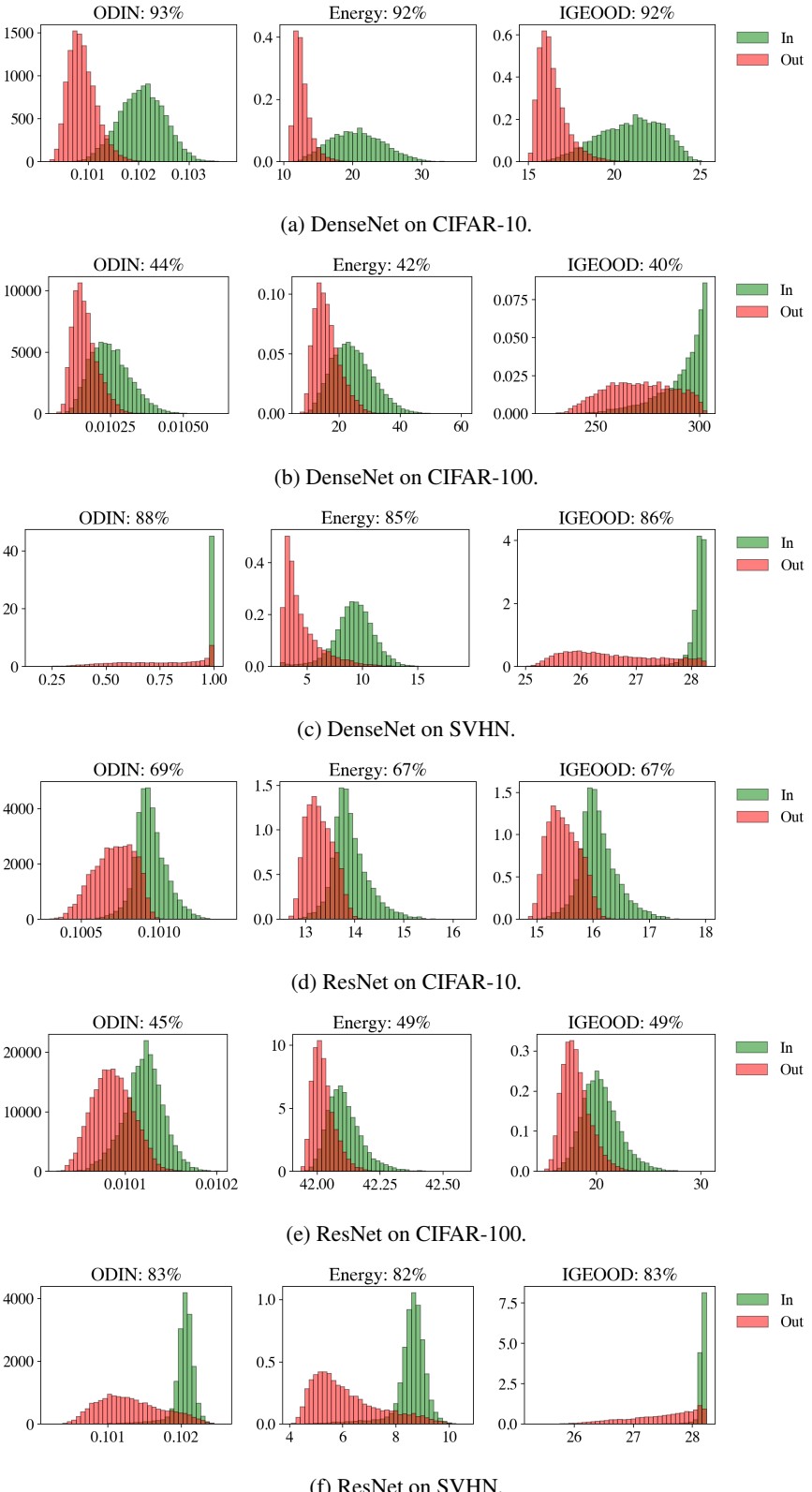

Figure 8: GREY-BOX setup. TinyImageNet as OOD dataset.

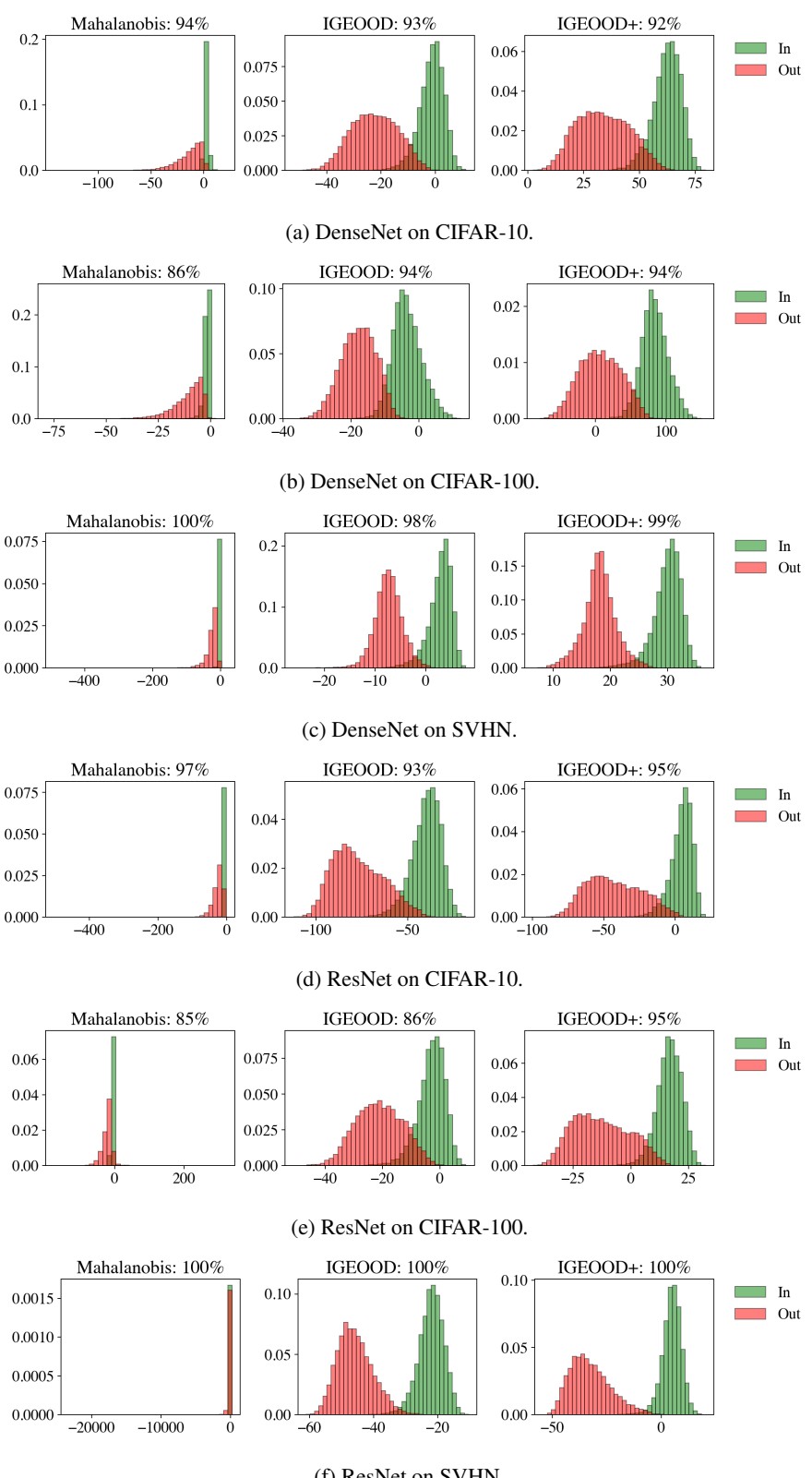

Figure 9: WHITE-BOX setup with adversarial data validation. TinyImageNet as OOD dataset.

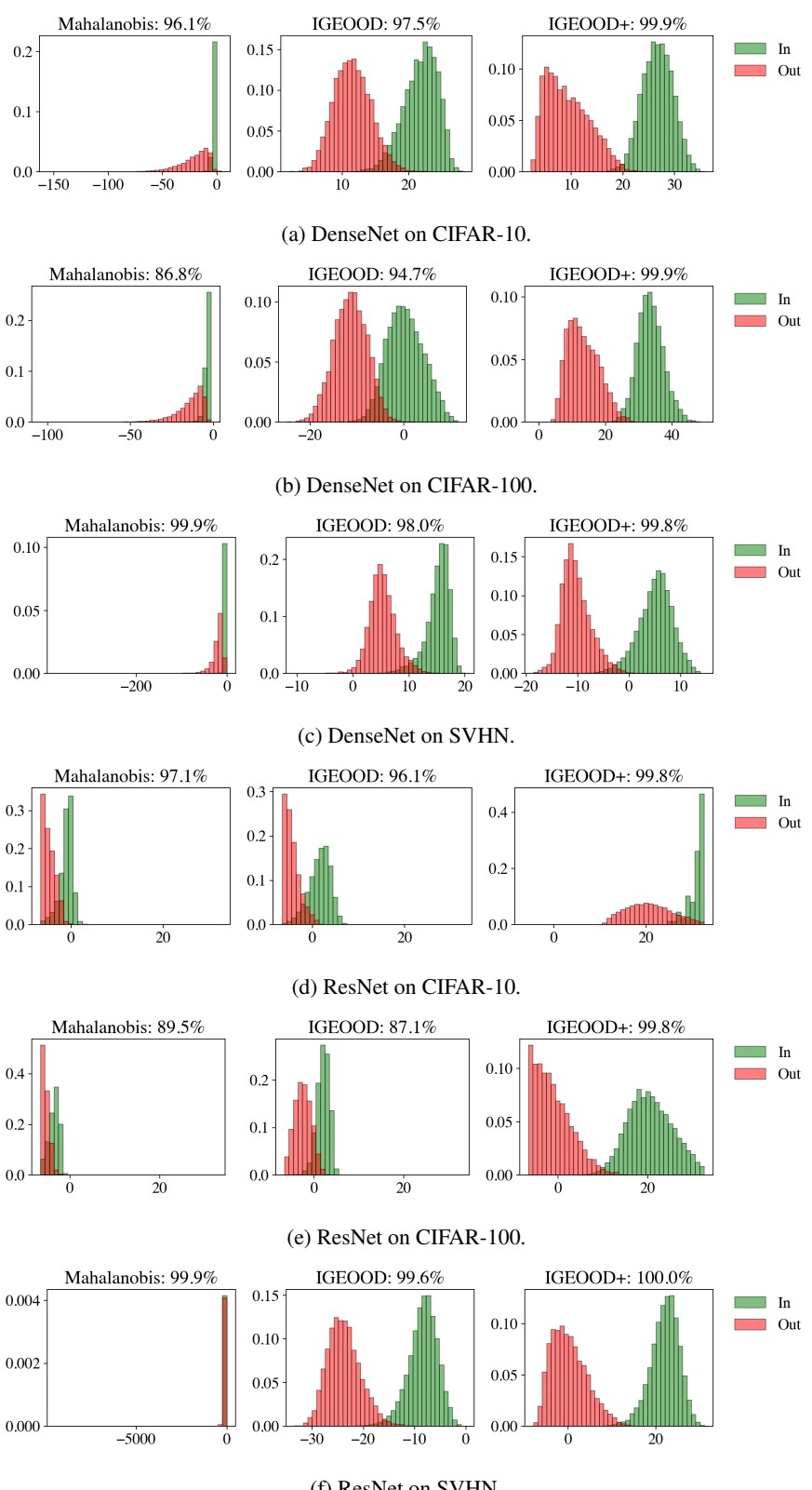

Figure 10: WHITE-BOX setup with validation on OOD data. TinyImageNet as OOD dataset.

