# OpenReview forum: "Igeood: An Information Geometry Approach to Out-of-Distribution Detection"
_ICLR.cc/2022/Conference — ICLR 2022 Poster_

### Official Review · Reviewer_dbMS · 2021-10-30

**Correctness:** 2
**Technical Novelty And Significance:** 3
**Empirical Novelty And Significance:** 2
**Recommendation:** 5
**Confidence:** 3

**Main Review:**

Strengths of the paper
1) The concept of applying the Fisher-Rao distance to out-of-distribution seems novel and important to me. Furthermore, examples on the Gaussian strengthen the motivation of the FIsher-Rao distance's theoretical benefit.
2) Proposed Fisher-Rao distance can be combined with pre-existing techniques, input-preprocessing, temperature scaling, and feature ensembling, to boost the OOD performance.
3) The proposed method shows competitive results on the black-box OOD detection setting.

Weakness of the paper
1) One major issue of the proposed algorithm is that we have to tune the parameter of centroid parameter of each class. While the authors have noted that they optimized for 100 epochs, I wonder how the extra computation time scales compared to the baselines. (e.g., Mahalanobis distance, energy-based distance), especially in the CIFAR-100 dataset, where we have to evaluate the mean of the centroid 100 times.
2) Furthermore, given the computation time of 1), experiment results are weak to champion IGEOOD as the OOD method suitable for pre-trained classifiers. In the grey-box setting, ODIN outperforms IGEOOD. Furthermore, in the white-box setting, the method is only compared against the Mahalanobis distance.

Comments


1. As mentioned in the weakness section, the wall clock time of the IGEOOD on the various datasets can help to solve the time efficiency issue of the method.


2. Furthermore, in the white-box setup, I suggest comparing the method not only against the Mahalanobis distance but also on the recently proposed methods (e.g., [1])


3. In figure 1(c), while it's convincing that the Fisher-Rao distance improves detection against type-2 OOD data, the two histograms are not normalized enough to make comparisons. I suggest matching the area of in-distribution data frequencies between two histograms.


4. ." Empirically, we show in the appendix (see Section C) that this confidence score does not degrade and
sometimes improves the in-distribution test classification accuracy".<<<< I found the results and the claim are rather redundant. If there is no stark improvement, why should we take a look into it?


5. For Figure 2, I suggest adding the distributions of the baselines for direct comparisons.


6. How the choice of validation dataset impacts performance <<< I am skeptical about the absolute, or relative robustness of IGEGOOD against the other algorithms. First, 8% variation in TNR does not seem so robust since the paper's gain against baselines is not bigger than 8%. Furthermore, I cannot find a major difference against baselines given the results in E.3.

==================================================================================================
References
[1] Detecting Out-of-Distribution Examples with Gram Matrices. C. S. Satry and S. OOre,


**Summary Of The Paper:**

UPDATE:

I acknowledge that I've read the author responses as well as the other reviews.

While the authors added further analysis on the proposed method, I am skeptical of the performance of the method since the proposed method requires validation OOD data to achieve SOTA performance and the runtime is much larger than MSP and energy baseline. However, I think the rebuttal clarified much of my concerns and therefore raise the score to 5 weak reject.

=================================================================

The paper proposes an out-of-distribution detection (OOD) metric based on Fisher-Rao distance which can be applied for pre-trained classifiers. First, the authors derive a Fisher-Rao distance applied to the distribution of softmax. Also, they motivate a toy example where the FIsher-Rao distance outperforms the conventional OOD metric, Mahalanobis distance. Furthermore, they formulate the Fisher-Rao distance-based framework, IGEOOD on Black(Grey)-Box, where we can only get access on the logit of the network output, and White-Box, where we can get access to intermediate feature layers. Finally, the authors compare IGEOOD against conventional OOD metrics on various out-of-distribution data and in-distribution data.

**Summary Of The Review:**

The paper proposed a new OOD detection framework, IGEGOOD. My biggest concern is that the experiment results are fairly weak given the complex procedure of obtaining the detection scores. Therefore, I am leaning towards the rejection of the paper.

---

> ### Author Response · Authors · 2021-11-21
> **Reply to reviewer dbMS**
>
> We thank the reviewer for their helpful comments. We will try to clarify these points.
>
> **About the weakness of the paper:**
>
> 1. We would like to clarify a misunderstanding.  The computation of the logits centroids is performed entirely offline based on two steps. In the first step, we save all the logits from the training set in memory and then run the optimization algorithm with the loaded data. Simulations accelerated by one GPU takes around 83s to save all logits and 1.1s/11s to obtain the centroids for CIFAR-10/CIFAR-100, respectively. We added to the appendix (see Section B.1 in the updated paper) a detailed study on the computation time for those steps involved in our algorithm.
>
> 2. For the Black-Box and Grey-Box settings, we achieve competing results compared to the current literature. ODIN performs usually better in the Grey-Box setting. In the White-Box setting, we improved the benchmark considerably compared to Mahalanobis, which is the most closely related method to IGEOOD. We also added to the appendix (see Section E.6) a table comparing IGEOOD to other methods that fit the White-Box setup, including reference [1]. We show improvement in all benchmarks.
>
> **Response to specific comments:**
>
> 1. We updated Figure 1(c) with normalized and aligned bins and ensured that the area of each histogram equals one.
>
> 2. The objective of this experiment is to verify if the proposed metric was aligned with the classification of samples, which is usually a good indicator for OOD detection performance. We agree this was an overstatement in the paper. We made the necessary corrections to the revised version.
>
> 3. We thank the reviewer for the suggestion. In the appendix (see Section B.3 and F), we added more detailed histograms of our metric in different network layers compared to the Mahalanobis baseline.
>
> 4. The experiments in Section E.3 of the previous version of the paper (now Section E.4) aim to show the consistency of IGEOOD in the choice of validation dataset. We agree that "robustness" might not have been the most convenient word. In the revised version, we updated the text accordingly. The table shows that, independently of the choice of validation dataset, IGEOOD provides competitive results to the literature in the Black-Box and Grey-Box settings. The range of around 8% TNR is also observed for ODIN and Energy methods.
>
>
> *We hope we have addressed most of your comments satisfactorily and kindly request you to revise your score.*

---

### Official Review · Reviewer_KjND · 2021-10-31

**Correctness:** 4
**Technical Novelty And Significance:** 3
**Empirical Novelty And Significance:** 4
**Recommendation:** 8
**Confidence:** 4

**Main Review:**

## Methodology

The authors propose two related methods: (1) measuring distance in the output space, and (2) measuring distance in the space of the hidden layer activations. (1) is quite intuitive, and as far as I know novel: to the best of my knowledge prior work typically considers just the confidence of the classifier, and not the full predictive distribution. (2) is quite similar to Mahalanobis [1], with the main difference being the distance metric used. However, the authors show significant improvements in performance compared to [1] in Section 4.3, justifying the proposed method.

## Results

The empirical results constitute the main strength of this paper: the method outperforms the considered baselines across the board. However, I have two concerns:

- The improvements in the Black-box setting appear very minor. In many cases, the method does not outperform the baselines, our improves the results by $O$(0.1%) (see e.g. the AUROC results for ResNet, in Table 1). It would potentially be helpful to add error bars to Table 1.
- The paper claims to set the new *state-of-the-art* on visual out-of-distribution detection, but the considered baselines are somewhat limited. OOD detection is a very active field, with many papers claiming to improve on the considered baselines [see e.g. 2-6]. From a quick comparison it seems like the results reported by the authors are competitive with the best methods I could find, but I think the authors need to do a better job comparing to prior work, in order to claim SOTA.

On the other hand, I want to highlight that the authors perform a fairly exhaustive experimental evaluation in terms of the out-of-distribution datasets considered for each in-distribution dataset, including both near- and far-OOD.

## Writing

The writing is generally clear, but there are several typos and minor inaccuracies.

## Comparison to other distance metrics?

As far as I understand, it is possible to replace the Fisher-Rao metric with any other distance metric in the method proposed by the authors? I think it would be interesting to see a comparison of the results with a few other standard metrics in the Black-Box set-up, e.g. KL-divergence, total-variation distance, etc. If the Fisher-Rao metric provides significantly better results, this experiment would strengthen the paper.


## References

[1] A Simple Unified Framework for Detecting Out-of-Distribution Samples and Adversarial Attacks
Kimin Lee, Kibok Lee, Honglak Lee, Jinwoo Shin

[2] Generalized ODIN: Detecting Out-of-distribution Image without Learning from Out-of-distribution Data
Yen-Chang Hsu, Yilin Shen, Hongxia Jin, Zsolt Kira

[3] Detecting Out-of-Distribution Examples with In-distribution Examples and Gram Matrices
Chandramouli Shama Sastry, Sageev Oore

[4] Hybrid Models for Open Set Recognition
Hongjie Zhang, Ang Li, Jie Guo, Yanwen Guo

[5] Deep Residual Flow for Out of Distribution Detection
Ev Zisselman, Aviv Tamar

[6] A Simple Fix to Mahalanobis Distance for Improving Near-OOD Detection
Jie Ren, Stanislav Fort, Jeremiah Liu, Abhijit Guha Roy, Shreyas Padhy, Balaji Lakshminarayanan

**Summary Of The Paper:**

The paper proposes a new group of methods for supervised OOD detection. In particular, the authors propose to use the Fisher-Rao distance between output distributions on the in-distribution data and test samples to detect OOD. The authors additionally propose to use Fisher-Rao distance in the hidden layer feature space, when possible (white-box setting). The method achieves strong empirical performance, improving upon standard baselines (such as Odin, Mahalanobis), especially in the white-box setting.

**Summary Of The Review:**

In summary, I recommend a weak accept for this paper. The empirical results are good, and the method is generally novel. I am open to increasing my score, if the authors address the concerns I raised in my review.

---

> ### Author Response · Authors · 2021-11-21
> **Reply to reviewer KjND**
>
> We thank the reviewer for their helpful comments.
>
> - *It would potentially be helpful to add error bars to Table 1.* We thank the reviewer for this suggestion. We added error bars based on the standard deviation to all of our main results. Please see the general reply above.
>
> - *From a quick comparison it seems like the results reported by the authors are competitive with the best methods I could find, but I think the authors need to do a better job compared to prior work, in order to claim SOTA.* To further compare our results in the white-box setting, we added a table where we compared IGEOOD to the original results from papers [1], [2], [3], [5], as they have a sufficiently large base for comparison. Unfortunately, the experiments are primarily oriented to far-OOD, as these benchmarks are most commonly used. Please refer to Section E.6 of the updated paper.
>
> - *About writing.* We have carefully checked the paper and made all necessary corrections.
>
> - *About comparison to other distance metrics.* We thank the reviewer for this suggestion. We added an empirical study for the use of the KL divergence in a Black-Box setting for OOD detection and how it is linked to the Fisher-Rao metric (see Section E.1 of the updated paper). By using the Fisher-Rao metric, we record an improvement of around 0.4% on average TNR compared to KL, which is considerable in our extensive Black-Box benchmark.
>
>
> *We hope we have addressed most of your comments satisfactorily and kindly request you to revise your score.*

---

> > ### Comment · Reviewer_KjND · 2021-11-29
> > **Thank you for the rebuttal**
> >
> > Dear authors, thank you for the rebuttal, including the new experiments and explanations. I am happy to increase my score given these new experiments and a more detailed comparison to prior work.

---

### Official Review · Reviewer_R2Ss · 2021-11-03

**Correctness:** 3
**Technical Novelty And Significance:** 3
**Empirical Novelty And Significance:** 2
**Recommendation:** 6
**Confidence:** 3

**Main Review:**

The theoretical basis of the proposed methodology is well described and the experiments appear to be well-designed overall. Empirically the proposed method outperforms other methods, especially in the white-box setting where OOD validation samples are available.

According to Table S8 (Table 8 in the Appendix), the performance difference of the different methods appears to be marginal. Can you provide statistical significance for the performance differences? In addition, according to Table S9 (grey-box), ODIN performs better than the proposed method in most cases. Can you add some discussion on why? Moreover, the current description of the results given in the main text is a bit misleading because these observations are not properly explained.

In the white-box setups, IGEOOD seems to perform best when OOD samples are available for validation (Table S10), but it does not when only the adversarially generated samples are used for validation(Table 2, Table S11). This result does not match the purpose of this study presented in Abstract and in Introduction (e.g., "IGEOOD applies to any pre-trained neural network, does not require OOD samples or assumptions on the OOD data").

Recently, many SSL (self-supervised learning)-based OOD detection methods have been developed. Examples are SSD[1] and CSI[2] that use the same confidence score as in Lee et al.[3] while utilizing self-supervision (simclr). I wonder 1) if the proposed method can be compared to these self-supervision-based models, and 2) whether the proposed geodesic distance can be applied to the SSL-based approaches.

[1] Sehwag, Vikash, Mung Chiang, and Prateek Mittal. "SSD: A Unified Framework for Self-Supervised Outlier Detection." International Conference on Learning Representations. 2020.
[2] Jihoon Tack, Sangwoo Mo, Jongheon Jeong, and Jinwoo Shin. Csi: Novelty detection via contrastive learning on distributionally shifted instances. Advances in Neural Information Processing Systems, 33, 2020.
[3] Lee, Kimin, et al. "A simple unified framework for detecting out-of-distribution samples and adversarial attacks." Advances in neural information processing systems 31 (2018).

The proposed model is based on the assumption that the layer output follows a Gaussian distribution. It is possible to provide validation or discussion of this assumption?

Regarding Eqs. (5) and (6), the authors note that taking the sum (5) instead of the minimum distance to the class conditional centroid produces better results. This looks interesting. Can you provide empirical results or more analysis on this?

It would be interesting to see the score distributions as shown in Figure 1(c) for the real datasets used in this study.

**Summary Of The Paper:**

This paper presents IGEOOD, a new method for detecting OOD samples by using geodesic (Fisher-Rao) distance in confidence scoring. It further combines confidence scores from the logit outputs and the layer-wise features of a deep neural network. The method is validated under various testing environments such as the availability of OOD data or the accessibility of latent features of a deep network. The idea of using Fisher-Rao distance for OOD detection seems novel and interesting.

**Summary Of The Review:**

This paper presents a novel idea of using Fisher-Rao distance for confidence scoring in OOD detection. The main idea is interesting, the paper is well structured in terms of methodological description and the problem/experimental setups, but the empirical results do not appear to be sufficient to validate the intended purpose of this study.

---

> ### Author Response · Authors · 2021-11-21
> **Reply to reviewer R2Ss**
>
> We thank the reviewer for their helpful comments.
>
> - *According to Table S8 (Table 8 in the Appendix), the performance difference of the different methods appear to be marginal. Can you provide statistical significance for the performance differences?* We thank the reviewer for this suggestion. We added error bars based on the standard deviation for our main results.
>
> - *According to Table S9 (grey-box), ODIN performs better than the proposed method in most cases. Can you add some discussion on why?* The average results show that ODIN outperforms our method by 0.9% on TNR (Grey-Box). We believe that given that the benchmark considered is very extensive, heterogeneous, including datasets considered far- and near-OOD, and with a standard deviation equal to 21% for ODIN and 20% for our method, the results are very competitive. For the Black-Box scenario, the reasoning is the same. IGEOOD shows an improvement of ~1% w.r.t other methods. Thus, in both cases, these small improvements are hidden by the standard deviation. What we can state, from observing Figures 5 and 6 in the updated paper, is that IGEOOD benefits more from temperature scaling than input pre-processing, while ODIN primarily benefits from input pre-processing. For the White-Box scenario, there aren't any doubts about the improvements of IGEOOD. The Black-Box and Grey-Box settings are included for the sake of completeness and to show that IGEOOD is still competitive in these other cases.
>
> - *Moreover, the current description of the results given in the main text is a bit misleading because these observations are not properly explained.* We thank the reviewer for pointing out this issue which was not clear enough. In the main text of the revised paper, we have made a more detailed statement about our results in the Black-Box scenario (see Section 4.2 in the updated paper).
>
> - *IGEOOD seems to perform best when OOD samples are available for validation (Table S10), but it does not when only the adversarially generated samples are used for validation(Table 2, Table S11).* In general, IGEOOD outperforms Lee et al. 2018 even for validation on adversarial data by improving TNR by large amounts in some cases (>10%), but being outperformed by smaller amounts (<4%) in other cases. Overall, IGEOOD still outperforms by 2.5% on average. In addition, we stated in the contributions that we could benefit from OOD samples in scenarios where a packet of OOD or adversarial samples is available. We also added this claim in the abstract to better reflect the observations.
>
> - *SSL (self-supervised learning)-based OOD detection methods:*
>     *1) If the proposed method can be compared to these self-supervision-based models.*  We believe that the frameworks of [1] and [2] are somewhat different from ours, as these propose learning representation to solve the problem of OOD detection by training a model from scratch. Thus, they cannot be directly compared to IGEOOD.
>     *2) Whether the proposed geodesic distance can be applied to the SSL-based approaches.* We thank the reviewer for this excellent suggestion. The proposed geodesic distance can be applied to these frameworks. It is a promising approach which will lead to future work.
>
> - *The proposed model is based on the assumption that the layer output follows a Gaussian distribution. Is it possible to provide validation or discussion of this assumption?* We thank the reviewer for their suggestion. Please refer to the general reply above, where we addressed this question.
>
> - *Regarding Eqs. (5) and (6), the authors note that taking the sum (5) instead of the minimum distance to the class conditional centroid produces better results. This looks interesting. Can you provide empirical results or more analysis on this?* We thank the reviewer for their suggestion. Please refer to the general reply above, where we addressed this question.
>
> - *It would be interesting to see the score distributions as shown in Figure 1(c) for the real datasets used in this study.* We thank the reviewer for the suggestion. We added the histograms for the scores for a few experiments in the appendix (see Figure 4 and Section F in the updated paper).
>
>
> *We hope we have addressed most of your comments satisfactorily and kindly request you to revise your score.*

---

> > ### Comment · Reviewer_R2Ss · 2021-11-29
> > **Thank you for the rebuttal**
> >
> > I appreciate the authors' efforts to respond and revise the manuscript. I still have some remaining concerns, but after reading all the other reviews and the author responses, I'm now more inclined to accept this paper. I will raise my score from 5 to 6.

---

### Official Review · Reviewer_eVhF · 2021-11-07

**Correctness:** 3
**Technical Novelty And Significance:** 3
**Empirical Novelty And Significance:** 3
**Recommendation:** 6
**Confidence:** 4

**Main Review:**

## Strengths & Weaknesses

**Clarity**: The paper is mostly clear and well written. The writing can be improved in some places (e.g. paragraph 2 in the Introduction)

**Novelty**: While the Fisher-Rao metric has been applied in the context of deep learning (natural gradient, regularization of training, etc. -- need references here), I've unaware of its use for anomaly/ out-of-distribution (OOD) detection. It's a reasonable extension to evaluate the utilization of Mahalanobis distance in the prior art (Hendycks and Gimpel, ICLR'17). The novelty meets the bar for a publication without rising to the level of being significant.

**Technically correctness**: The material is technical correct in the large. I went through the approach including the math in the paper and skimmed through the Appendix. I'm mostly familiar with the topic and the material seems correct though I didn’t check the appendix comprehensively.

*(TC 1) Gaussianity*: There are two main parts for the proposed score – one based on the SoftMax output – this seems ok. My main concern is with the other part -- using the multivariate Gaussian model with a diagonal covariance for the feature spaces without first validating the premise with a Gaussianity test. While testing for high-dimensional multivariate Gaussians may pose difficulties, the diagonality assumption should make this feasible. I recommend that evaluation and analysis of this assumption is added to the paper.

*(TC 2) PMFs and PDFs*: while the SoftMax formulation provides a posterior distribution in the label space, having this interpretation for the feature layers for a single input sample without further grounding in theory/ past literature makes the approach ad hoc. I'd like to see authors clarify this.

*(TC3) Max and Average*: The authors comment that (for the SoftMax output) they obtain slightly better results using (6) -- average() rather than (5) -- min() over the classes.

(a) Would using the average be tantamount to computing the probability with respect to a mixture distribution and admit a model more applicable for the scenario (single sample case) than a PMF/ PDF estimate interpretation?

(b) Was this also tried with the features corresponding to equations (12) and (13)?

**Experimental Evaluation**: A good number of experiments have been conducted and presented in the main paper for the black-box (Table 1) and the white-box (Table 2) scenarios. These are further supported by additional experiments in the Appendix E (Tables 5-11) as well as ablation studies for various (hyper)parameters important to the proposed approach. This is very good.

On the flip side, some parts of the experimental validation can be improved to support better the central claims – that OOD scores based on the Fisher-Rao information metrics outperform the previous state of the art.

*(EE1) Difficulty of directly comparing with published results*: Since the experimental settings seem different from other papers (DNN models used, finetuning dataset, etc.), it is really hard to directly compare tables in the paper to those published in the compared SOTA. I tried to do this both for the Tables in the main paper as well as in the Appendix. Since this beats the due diligence review process, the authors should explain why this is ok.

(a) Black-Box settings: Entries in Table 1 (and Table 8) can't be compared with Table 2 in Hendryks and Gimpel (ICLR 2017) referred to as Baseline, Table 1 or Table 2 in Liu et al (NeurIPS 2020) or with ODIN - Liang et al (ICLR 2018).

(b) White-Box settings:  I compared some numbers across Tables 10 and 11 with Table 2 in Lee et al (NIPS 2018) -- discrepancies in results exist, sometimes very significant,  but mostly (not all) the comparative accuracy improvements hold.

I gave up on a more comprehensive cross-checking of the results but since there are differences in the setups.

*(EE2) SOTA used for comparison*: Apart from Liu et al (NeurIPS 2020), all the compared approaches seem 3-4 years old. The authors should respond on whether those results are SOTA.

*(EE3) Inconclusive evidence for improved performance*: Performance improvements vis-a-vis the reported SOTA is mixed – ref. Tables 8 – 11 though I consider the results to be promising. The authors should discuss this in their response.

**Reproducibility**: I expect the results to be reproducible since enough details are shared in the paper and the code has been made publicly available at https://github.com/igeood/Igeood.



**Summary Of The Paper:**

The paper proposes to use a score based on the Fisher-Rao information metric (the Riemannian metric in the space of probability distributions) for the detection of out-of-distribution samples input to a trained DNN. While the output SoftMax probabilities are used in the black-box and grey-box scenarios, the learnt features in the intervening DNN layers are additionally used in the white-box scenario. The approach models each sample as providing posterior probabilities – (a) the SoftMax probability in the label space, and (b) class-conditional PDFs over the corresponding feature spaces for each DNN layer. These latter are modeled as multivariate Gaussian distributions with diagonal covariance matrices. Extensive experiments on existing benchmarks are conducted for comparative results against the state of the art demonstrating promising results.

**Summary Of The Review:**

I recommend to accept the paper.

It is a good addition to the set of methods on an important topic - OOD. The approach is novel, reasonably principled and the code has been made publicly available at [https://github.com/igeood/Igeood](https://github.com/igeood/Igeood) which is good for the community to able to put the above methods to test.  The validation is comprehensive though the results are somewhat inconclusive though promising. There are some concerns regarding the validation of assumptions and the experimental evaluation. My preliminary assessment is that the submission passes the criteria for acceptance to this venue.

---

> ### Author Response · Authors · 2021-11-21
> **Reply to reviewer eVhF**
>
> ​​We thank the reviewer for their helpful comments.
>
> **Technically correctness:**
> - (TC1) We thank the reviewer for their suggestion. Please refer to the general reply above where we addressed this question.
> - (TC2) According to the Test of Gaussianity conducted in Section B.3, the probability distribution of the different hidden layers across the network can be reasonably modelled as different multivariate Gaussian distributions. This assumption induces a class of distributions for which we can compute distances between two of the members  (e.g., between two distributions with different statistics such as the means and/or covariance matrices). Thus, the main idea is the same as for the last layer (i.e., the softmax probability), which consists of measuring the Fisher-Rao distance between two probability distributions, i.e. the in-distribution reference and the test distribution that we are trying to compare to each other, within the class of multivariate Gaussian distributions.
> - (TC3) **(a) Average vs. minimal distance scores.** To further investigate the effects of taking the average instead of the minimal distance between the soft-probabilities, we added an ablation study between them (see Section E.1). However, modelling the underlying distributions of the hidden layers as a mixture of Gaussian distributions would require computing the Fisher-Rao distance between mixtures of Gaussian distributions. Although this approach would be more general, unfortunately, to the best of our knowledge, there is no known expression for the Fisher-Rao distance between mixtures of Gaussian distributions. This requires solving a rather complex problem which is out of the scope of our paper but would be an interesting extension for future work.
>
>     **(b) Sum in Eqs. (12) and (13) vs. min distance.** We ran experiments for the sum in equations (12) and (13), but on average we obtained worse results. Thus, we did not report them in the updated paper.
>
> **Experimental Evaluation:**
> - (EE1) We would like to clarify that the DNNs and general setup used are the same as those in seminal and recent works: Lee et al. 2018, Liang et al. 2018 and Liu et al. 2020. However, slight changes were made:
>     1. The setup in Liang et al. 2018 and Liu et al. 2020 is equivalent to our Grey-Box setup, where input-preprocessing is allowed. Thus, Table 2 of Liang et al. 2018 is comparable to the full Table 13 in the updated paper. We agree with the reviewer regarding the fact that the Black-Box results cannot be compared to the results of these papers, as Energy and ODIN do not report table results for the Black-Box setup (i.e., their methods implemented without input pre-processing). When comparing the results of Table 13 to the ones found in Liang et al. 2018, we obtained a minor difference, because we recalculated the statistics of the training data for data transformation. With the updated statistics, the classification model accuracy is increased by 0.01% on CIFAR-10 and decreased by 0.01% on CIFAR-100. For CIFAR-10, ODIN reports 7.5% and 3.8% FPR on TinyImageNet and LSUN resized, respectively, while in our setup we obtained 7.4% and 3.8%. For CIFAR-100, they report 57% and 58% FPR on TinyImageNet and LSUN resized, respectively, while we obtained 56.2% and 57.8% with their method. Since the only modification is the data statistics, we believe this explains the difference in those results. So, in general, we slightly improved their results.
>     2. For Mahalanobis, the statistics of the training data were also recomputed. Also, we partition the adversarial set by simply taking the first 1000 samples, while Lee et al. 2018 partition the data differently. If we compare their results in terms of TNR (Table 2 in their paper) with the results we obtained with their method in our setup (in parenthesis), we obtain:
>     ```
> DenseNet/CIFAR10:
> TinyImageNet: 95.0 (96.9) | 94.9 (93.9)
> LSUN: 97.2 (98.2) | 97.2 (96.3)
> SVHN: 90.8 (90.1) | 89.6 (87.0)
> ```
>     For our setup, these results correspond to Tables 14 and 15. The differences are on average minors.
> - (EE2) We compared our results with more recent literature, which was also suggested by other reviewers. The results are available in Section E.6.
> - (EE3) We brought an extensive and varied benchmark to analyze if the OOD detection methods are consistent across different datasets and models. We show competitive results for the Black-Box and Grey-Box settings, but not conclusive enough to claim an improvement of the SOTA. We modified the text in the contribution part accordingly. For the White-Box, we are superior in some cases to Mahalanobis for validating with adversarial data, bringing an improvement of 2.5% in terms of the average TNR. In addition, our method can also benefit from OOD samples, achieving SOTA. We think that the comparison in Section E.6 contributes to supporting our claim.
>
> *We hope we have addressed most of your comments satisfactorily and kindly request you to revise your score.*

---

> > ### Comment · Reviewer_eVhF · 2021-11-28
> > **Response to the Review**
> >
> > I sincerely thank the authors for their thoughtful engagement with the review and the detailed revisions/ additional experiments. While I don't exactly agree with authors' on their some of their responses (TC2, for example), I'm satisfied with the others for the purpose of this reviewing exercise.
> >
> > I note that concerns regarding clear empirical benefits over SOTA still remain in some settings (though I consider the methodology and tools used to be important enough to serve as the basis for further improvements).
> >
> > At this point, I'm recommending to accept the paper while retaining my rating (though I'm inclined to bump it up). I will finalize it as the conversation with the other reviewers unfolds.

---

### Author Response · Authors · 2021-11-21
**Summary of the main changes in the paper**

We thank the reviewers for their very helpful comments and suggestions. The paper has been carefully revised by taking into account all reviewer’s comments. The main changes are listed below:

- **Improving the performance of  IGEOOD.** We have further improved IGEOOD by introducing a small modification to eq. (13) which consists of also calculating the empirical mean of the available OOD samples hidden representation to better model the Gaussian reference distribution.
- **Additional numerical results.** We added additional comparisons with the state-of-the-art methods, according to the reviewers request (see Section E.6 in the updated paper).
Average vs. minimal distance scores. Two out of four reviewers requested to further discuss why we calculated the average  of the distances instead of the minimal distance. To better understand our motivation for this choice, we included in Section E.1 an empirical study comparing the results of both metrics.
- **Test of Gaussianity.** Two out of four reviewers raised questions about the validity of the Gaussianity assumption in modelling the distribution of the latent features. In order to seriously study this point, according to the reviewers suggestion, we added in the appendix (see Section B.3) a feature-wise test of Gaussianity. More precisely, we calculated the *Shapiro-Wilk test* of Gaussianity for each coordinate of every hidden feature. We averaged the resulting  *W statistics* results across all coordinates. The corresponding results show that the assumption of Gaussianity holds true for the outputs of the first convolutional layer, i.e. Block 1 and Block 2. Whereas, we observed a small decline in the test statistics for the penultimate layer output for which the distributions showed to be more peaky and less symmetric. The outcome of this experiment is consistent with recent works in the literature which have motivated empirically (see Lee et al. 2018) that samples drawn from different classes belong to clusters in the hidden representations of a DNN.
- **Error bars for numerical results.** Two out of four reviewers requested to introduce error bars and statistical confidence in our results. To this end, we added error bars based on the standard deviation for our main results.

---

### Author Response · Authors · 2021-11-30
**We acknowledge all reviewers for their time and thoughtful comments**

We want to acknowledge the reviewers for their thoughtful and valuable comments and efforts towards improving our manuscript. We believe the discussion period was very useful to truly revise and improve the presentation of our results.

---

### Decision · Program_Chairs · 2022-01-20

**Decision:**

Accept (Poster)

**Comment:**

This paper introduces a novel approach for out of distribution detection that generates scores from a trained DNN model by using the Fisher-Rao distance between the feature distributions of a given input sample at the logit layer and the lower layers of the model and the  corresponding mean feature distributions over the training data.

The use of Fisher-Rao distance is novel in the context of OOD, and the empirical evaluations are extensive.  The main concerns of the reviewers were the limitations of the Gaussianity assumption used in computing the Fisher-Rao distance and the use of the sum of the Fisher-Rao distances to the class-conditional distributions of the target classes rather than the minimum distance. These concerns were addressed satisfactorily in a revision. In terms of technical novelty, experimental evaluation and novelty, the paper is above the bar of acceptance.